# Rational Design 2-Hydroxypropylphosphonium Salts as Cancer Cell Mitochondria-Targeted Vectors: Synthesis, Structure, and Biological Properties

**DOI:** 10.3390/molecules26216350

**Published:** 2021-10-20

**Authors:** Vladimir F. Mironov, Andrey V. Nemtarev, Olga V. Tsepaeva, Mudaris N. Dimukhametov, Igor A. Litvinov, Alexandra D. Voloshina, Tatiana N. Pashirova, Eugenii A. Titov, Anna P. Lyubina, Syumbelya K. Amerhanova, Aidar T. Gubaidullin, Daut R. Islamov

**Affiliations:** 1Arbuzov Institute of Organic and Physical Chemistry, FRC Kazan Scientific Center of RAS, 8 Arbuzov St., 420088 Kazan, Russia; a.nemtarev@mail.ru (A.V.N.); tsepaeva@iopc.ru (O.V.T.); mudaris@iopc.ru (M.N.D.); litvinov@iopc.ru (I.A.L.); sobaka-1968@mail.ru (A.D.V.); tatyana_pashirova@mail.ru (T.N.P.); aplyubina@gmail.com (A.P.L.); syumbelya07@mail.ru (S.K.A.); aidar@iopc.ru (A.T.G.); daut1989@mail.ru (D.R.I.); 2Alexander Butlerov Institute of Chemistry, Kazan (Volga Region) Federal University, 18 Kremlevskaya St., 420008 Kazan, Russia; narutospace@icloud.com

**Keywords:** phosphonium salt, P–C bond formation, oxirane, glycidyl ether, triphenylphosphine, addition reaction, nucleophilic substitution

## Abstract

It has been shown for a wide range of epoxy compounds that their interaction with triphenylphosphonium triflate occurs with a high chemoselectivity and leads to the formation of (2-hydroxypropyl)triphenylphosphonium triflates **3** substituted in the 3-position with an alkoxy, alkylcarboxyl group, or halogen, which were isolated in a high yield. Using the methodology for the disclosure of epichlorohydrin with alcohols in the presence of boron trifluoride etherate, followed by the substitution of iodine for chlorine and treatment with triphenylphosphine, 2-hydroxypropyltriphenylphosphonium iodides **4** were also obtained. The molecular and supramolecular structure of the obtained phosphonium salts was established, and their high antitumor activity was revealed in relation to duodenal adenocarcinoma. The formation of liposomal systems based on phosphonium salt **3** and *L*-α-phosphatidylcholine (PC) was employed for improving the bioavailability and reducing the toxicity. They were produced by the thin film rehydration method and exhibited cytotoxic properties. This rational design of phosphonium salts **3** and **4** has promising potential of new vectors for targeted delivery into mitochondria of tumor cells.

## 1. Introduction

Phosphonium salts (QPSs) are organoelement compounds of importance in chemistry, pharmacology, biochemistry, etc. [1,2,3,4]. QPSs are used as organocatalysts in the reactions of C–C bond formation, annulation, etc., as a Lewis acid in organic synthesis [5].

Penetration of QPSs across biological membranes is one of the important properties of these compounds. Their high hydrophobicity and delocalized positive charge on the phosphorus atom lead to the tendency to accumulate in areas with a high membrane potential. Therefore, QPSs are actively used for the mitochondria targeted drug delivery [6,7,8,9,10,11]. Mitochondrial dysfunction is one of the key pathogenetic links in many diseases, such as metabolic syndrome, and cardiovascular and neurological disorders [12]. In this respect, in recent decades, the mitochondria-oriented approach has been intensively studied for the treatment of metabolic and degenerative diseases [13,14,15,16,17,18,19,20]. QPSs play a key role in implementation of this approach. Moreover, anti-cancer drug functionalized with triarylphosphonium groups provides an expanding spectrum of biological activity and shows effectiveness in the case of the resistant cells and pathogenic microorganisms [21]. This is clearly demonstrated by numerous examples: QPS-derivatives of chlorambucil [22], paclitaxel [23], doxorubicin [24], and metformin [25]. Incorporation of QPS into antibiotic molecules such as ciprofloxacin [26,27], azithromycin [28,29], doxycycline [28,29] significantly improves the antibacterial activity of doxycycline against Gram-positive bacteria. Azithromycin and doxycycline can also be considered as anti-tumor agents.

It is important to note that phosphonium groups should be considered not only as vector fragments in drug conjugations, but also as a pharmacophore [30,31]. This is especially shown in inhibition of respiratory chain complexes [32,33], Krebs cycle enzymes [34].

Creation of effective targeting drug delivery systems to organs and tissues is a topic of medicinal chemistry and pharmacology. [35]. QPS-modified delivery systems exhibit improved cellular uptake and selective targeting to mitochondria. This leads to inhibition of P-glycoprotein and suppression of drug resistance and cancer metastasis. [36]. Similar actions were obtained with QPS-modified glycol-chitosan polymer microspheres [37] and QPS-functionalized epigallocatechin gallate capped gold nanoparticles [38].

The functionalized QPS-target compound’s efficiency largely depends on the structure, in particular on the functional environment. Undoubtedly, identification of the new vector systems containing a linker part for the lipophilicity correction and conformational rigidity of molecule is urgent. The most widespread conjugate formation is the linker approach supposing a spatial differentiation of the lipophilic triarylphosphonium group and an active moiety of conjugate. To implement this approach, synthesis of the functionally substituted QPSs bearing carboxyl, hydroxyl, and amino groups as an additional function was developed (Figure 1). Due to the presence of these groups, functionally substituted QPSs can be easily conjugated to the corresponding functional groups of target drugs (hydroxyl and carboxyl) since esterification reactions and amide bond formation are the most developed processes in organic chemistry. This approach is the most gentle and rational way. The second approach is the esterification of drugs containing a carboxyl group with terminal bromo (iodine) alkanols with follow reaction with triphenylphosphine (Figure 2). This approach is less convenient, especially when the parent drug contains other functional groups that can react with triphenylphosphine.

Currently, the most commonly used functionally substituted QPSs contain compounds with a carboxyl **(I)**, hydroxyl **(II)**, or amino **(III)** groups in the alkyl substituent of phosphorus atom (Figure 3).

The most common method for the QPSs **(I)** synthesis is alkylation of triarylphosphines with halogenated carboxylic acids (Figure 4a). This reaction is carried out both without a solvent and in various mediums (THF, benzene, xylene, acetonitrile, etc.). QPSs were obtained using this method from triphenylphosphine for n = 1 [39,40], 2 [39,40,41,42,43,44,45,46,47,48,49,50,51,52,53,54,55,56,57,58,59], 3 [39,40,45,47,60,61,62,63,64,65,66,67,68,69,70], 4 [40,44,47,62,65,71,72,73,74,75,76,77,78,79,80,81,82,83,84], 5 [45,55,61,62,65,67,85,86,87,88,89,90,91,92,93,94,95,96,97,98,99,100,101,102], 6 [44,47,50,58,62,87,103,104,105,106,107,108,109,110,111,112,113,114], 7 [47,62,86,115,116,117,118,119,120,121,122], 8 [44,55,62,89,120,123,124,125,126,127,128], 9 [62,65,72,86,126,129,130,131], 10 [45,55,62,68,109,125,129,132,133,134,135], 11 [45,47,86,89,119], 12 [45,133], 15 [136]. Salts for n = 2 were obtained from tri(4-methoxyphenyl)phosphine, tri(2,4,6-trimethoxyphenyl)phosphine, and bromopropionic acid [137,138]. The second approach is the addition of triphenylphosphine to unsaturated carboxylic acids (Figure 4b) followed by treatment with gaseous or concentrated HCl or HBr [139,140]. It should be mentioned the reactions of lactones with triphenylphosphine in the presence of hydrohalic acid (Figure 4a) [141,142,143].

QPSs containing a hydroxyl group **(II)** are mainly obtained by the reaction of terminal halogen alkanols with triphenylphosphine (Figure 4c) (for an overview, see [144]). The corresponding bromine(iodine) alkanols are usually heated with triphenylphosphine in acetonitrile. This method was used to obtain phosphonium salts for n = 2 [47,145], 3 [40,47,100,146,147], 4 [47], 5 [47,148], 6 [47,149,150,151,152,153,154], 7 [148,155], 8 [148,156,157], 9 [156,158,159,160], 10 [156,157,161,162,163,164,165], 11 [156,157,162,165], 12 [151,157,159], 13 [166], 14 [167].

Phosphonium salts with amino group **(III)** were obtained by reactions of corresponding terminal bromoamines with triarylphosphines, for example, for n = 2 [40,47], 3 [54,142].

This work proposes an efficient and versatile approach for the preparation of 2-hydroxypropylphosphonium salts containing various additional functional substituents (halogens, alkoxy, acyloxy). The approach is based on reaction of oxiranes with triphenylphosphonium triflate. These compounds display anti-tumor activity. To improve the bioavailability and cellular uptake of obtained functionalized phosphonium salts, the nanotechnology-based approach was developed. For this purpose, phosphonium salt-modified liposomal delivery systems based on L-α-phosphatidylcholine were prepared. The incorporation of phosphorus-containing pharmacophore/vector amphiphils into liposomal membrane provides an ability to trap drugs effectively and prevent their leakage as well as rapid release from liposomes.

## 2. Results and Discussion

### 2.1. Chemistry

#### 2.1.1. Reaction of Triphenylphosphonium Triflate with Halomethyloxiranes

As a rule, the interaction of oxiranes with triarylphosphines is carried out under hard conditions and often leads to the dimerization and partial polymerization of oxirane. Thus, the reaction of triphenylphosphine with epichlorohydrin, recently studied in [168], leads to the oxirane dimerization into a dioxane structure ((1,4-dioxane-2,5-diyl)-bis(methylene))-bis(triphenylphosphonium)chloride. The opening of oxirane ring with the formation of triaryl (2-hydroxyalkyl) phosphonium salts occurs in the presence of acids in neutral solvents (alcohols, dichloromethane) [169,170,171,172,173] or during the triarylphosphine reaction in phenol used as the solvent [174].

In our work, this approach was applied to a wide range of new oxiranes using the most convenient and stable triphenylphosphonium triflate (**1**). This compound was easily obtained by mixing triphenylphosphine with trifluoromethanesulfonic acid in dichloromethane [170,173,175]. The signal in the ^31^P NMR spectrum of this compound strongly depends on the ratio of reagents. The signal of phosphonium salt **1** appears as a broadened singlet with a slight excess of triphenylphosphine. This signal shifts towards strong fields with an increase in the triphenylphosphine proportion. This is due to the exchange between triphenylphosphine and salt **1**. The salt signal appears as a doublet (δ_P_ 4.0 ppm, ^1^*J*_HP_ 532 Hz) at the exact ratio of the starting compounds [175].

The reaction of halomethyloxiranes **2a**–**e** with compound **1** occurs under mild conditions (−10 °C) with regioselective opening of three-membered ring at the O^1^–C^3^ bond in accordance with Krasusky’s rule for opening epoxides under neutral conditions [176,177,178] and the formation of triphenylphosphonium triflates **3a**–**e**. Despite protonation of oxygen, which often leads to initial opening of oxirane at the O^1^–C^2^ bond and carbocation formation (S_N_1 mechanism). In our case, the mechanism of S_N_2 nucleophilic substitution at the C^3^ atom was realized precisely in the protonated form of oxirane (structure A, Figure 5). This is consistent with the data of work [174]: The inversion of the configuration of corresponding chiral carbon atom (C^3^) proceeds under the action of triphenylphosphine in neutral conditions [176,177,178].

The use of optically active chloromethyloxiranes **2e** (*R* and *S*) in this reaction led to the formation of salts **3e** with the retained configuration of the C^2^ atom (*R* and *S*). Even in a strongly acidic environment, the nucleophilic attack of triphenylphosphine on the C^2^ atom is more preferable than the preliminary electrophilic opening of oxirane. QPS structures were confirmed by NMR, IR, and mass spectrometry. The characteristic values of chemical shifts and spin–spin coupling constants for sp^2^-carbons of phenyl substituents at phosphorus are observed in the ^13^C NMR spectra of salts **3**, completely consistent with QPS structures [179]. The structure of all compounds, including optically active derivatives **3e**, ***R*** and **3e**, ***S*** was proven by XRD. Chiral derivatives **3e**, ***R*** and **3e**, ***S*** form enantiomeric crystals in the polar space group P2_1_ with two independent molecules (formula units). Since the conformation of independent cations and anions is the same, therefore, one independent pair of anion–cation is shown in Figure 1.

An unexpected result was obtained in the study of **3a** crystals synthesized from racemic epichlorohydrin. As in crystals of optically active compounds, racemic product **3a** was crystallized in the same polar space group P2_1_ with two independent molecules A and B (formula units). In this case, crystal cell parameters of **3a** are different from those of crystals **3e**, ***R*** and **3e**, ***S***. The hydroxyl substituent at the C^2^ atom in one of independent molecules was disordered over two positions (Figure 2) with almost equal occupancy (0.48 and 0.52, respectively). This means that enantiomeric molecules with both *R*- and *S*-configurations of the C^2^ atom were localized in the position of an independent molecule B.

Crystals of **3a** are non-racemic. They have a ratio of enantiomeric molecules equal to 3:1. Since the substance is racemic in the mass, it should contain an equal ratio of isostructural enantiomeric crystals [*R*/*S* = 3/1 и *R*/*S* = 1/3]. Crystals of this type are called anomalous conglomerates.

Racemic compounds **3b**–**d** crystallize in isostructural centrosymmetric crystals in the space group P2_1_/n with one independent formula unit. The molecular geometry of **3b**–**d** are shown in Appendix A.

In this case, the unit cell volume of the crystals of this group has a similar value. It should be noted that the hydrogen bonds of hydroxyl group in enantiopure crystals **3e** are formed with two oxygen atoms of triflate anion. One of independent molecules forms H-bonds with two oxygen atoms and one fluorine atom in crystal **3a** and the second molecule (with a disordered hydroxyl group) forms with only one oxygen atom (Figure 2). H-bonds are formed only with one oxygen atom and one fluorine atom in centrosymmetric crystals **3b**–**d**. Hydrogen bonds are formed only between the anion and the cation (0D-motive of hydrogen bonds) in both centrosymmetric and noncentrosymmetric crystals. Perhaps a formation of hydrogen bonds of cation with various atoms of anion leads to some differences in the compound packing of this group in crystals. The result is the formation of centrosymmetric and non-centrosymmetric crystals.

The molecule geometric parameters of **3a**–**d** are the same within experimental errors and correspond to the bond lengths and angles in similar organic compound fragments and triflate anion. Crystal packing of this group is given by dispersion interactions, while π–stacking interactions between benzene rings were not observed.

#### 2.1.2. Reaction of Triphenylphosphonium Triflate with Alkyl- and Acyl Glycidyl Ether 

The reaction of alkyl- and acylglycidyl ethers **2f**–**l** with triphenylphosphonium triflate was carried out under mild conditions, with a high regioselectivity, and led to the formation of 3-alkoxy(acyloxy)-2-hydroxypropyltriphenylphosphonium triflates **3f**–**l** (Figure 6) with almost quantitative yields.

The structures of all compounds were confirmed by NMR, IR, and mass spectrometry. Compounds **3h**, **k**, **l** are thick oils, and other phosphonium salts are crystalline substances. The structures of salts **3f**, **g**, **i**, **j** were proven by XRD. The molecule geometries in the crystal are shown in Appendix A. The phosphorus atom has a distorted tetrahedral configuration both in salt molecules **3a–e** and in compounds **3f**, **g**, **i**, **j**. Unlike the former, the crystals of racemic compounds **3f**, **g**, **i**, **j** are not isostructural. Crystals **3g** and **3h** are non-centrosymmetric, and their space groups are nonpolar (Pn and Cc). Taking into account unit cells parameters of **3f** and **3g** it was assumed that they are isostructural. The methoxy derivative **3f** crystallizes in the centrosymmetric space group P2_1_/n. Compound **3g** crystallizes in the Pn group with two independent molecules per unit cell.

For crystals of all these compounds, hydrogen bonds are formed only between the anion and the cation (0D-motive hydrogen bonds).

#### 2.1.3. Synthesis of 3-Alkoxy(Iodo)-2-hydroxypropyl Triphenylphosphonium Iodides

To obtain 3-alkoxy-2-hydroxypropyltriphenylphosphonium iodides, a two-stage approach was applied starting from epichlorohydrin (Figure 7). Epichlorohydrin was disclosed with the corresponding alcohol in the presence of boron trifluoride etherate, according to [180]. The obtained 3-alkoxy-2-hydroxy-1-chloropropanes **5** were purified by distillation in vacuo. Then the Finkelstein reaction was carried out with sodium iodide in acetonitrile in the presence of dibenzo-18-crown-6 as a catalyst. The process was monitored by TLC and ^1^H NMR. Target phosphonium salts **4** were formed by heating the obtained 3-alkoxy-2-hydroxy-1-iodopropanes **6** with triphenylphosphine in acetonitrile in high yields.

The structure of compounds **4a**–**d** was proven by NMR and IR, and the structure of phosphonium salt **4a** also by XRD. Molecule geometry **4a** in the crystal is shown in Appendix A. It is interesting to note that Finkelstein’s reaction with compound **3a** led to the formation of two phosphonium salts—2-hydroxy-3-iodotriphenylphosphonium triflate **3d** and iodide **7** (Figure 8). The obtained salts were isolated and purified by colomn chromqatography. The structure of salt **7** was proven by NMR and XRD (Appendix A).

Some regularities in the spatial structure of investigated 12 compounds were formulated. The conformation and geometry of isopropanoltriphenylphosphonium cations in crystals is the same. The propanol fragment has a transoidal conformation (the P–C^1^–C^2^–C^3^ torsion angles are close to 180°). Hydroxyl groups of cations form hydrogen bonds with triflate or iodide anions. Closed 0D-hydrogen bond motifs were formed. Crystals of studied compounds are formed due to conventional dispersion interactions, while π–stacking interactions between benzene rings were not observed. All interplanar distances were more than 3.2 Å. Distances between the benzene rings were more than 5 Å. Bi- and trifurcate hydrogen bonds with two oxygen atoms and one fluorine atom of anion were observed in crystals with triflate anions. The increase in the ellipsoid size of anisotropic displacements of oxygen atoms can be explained by the “switching” of hydrogen bond from one oxygen atom to another (disorder of hydrogen bond).

#### 2.1.4. Liposomal Systems Based on Amphiphilic Triflates of Acyloxypropylphosphonium and L-α-Phosphatidylcholine

The creation of drug and gene delivery nanosystems is one of innovative approaches in cancer treatment. Liposomal systems have been applied in clinical studies [181,182]. Since the first liposomal product Doxil^®^, anticancer drugs such as DaunoXome^®^, Depocyt^®^, Myocet^®^, Mepact^®^, Marqibo^®^, and Onivyde™ have been successfully created [183,184,185]. The range of liposomal preparations is constantly expanding. Nanocarriers improve bioavailability and ensure targeted drug transport, selective targeting of cancer cells, and controlled drug release [186]. To improve the stability of the liposome systems and to protect against capture by the immune system cells, liposomes were modified by the ionic or non-ionic surfactants and/or polymers incorporation into phospholipid membrane [187]. 

Liposomes were modified with synthetic amphiphiles **3k** and **3l**. Characteristics of obtained liposomal formulations—hydrodynamic diameter (Zaverage, (nm)), zeta potential (Z, mV), polydispersity index (PdI), encapsulation efficiency (EE, %), and loading capacity (LC, %)—are presented in Table 1. The fluorescent dye Rhodamine B was selected as a model drug.

The liposome size was about 100 nm, and the zeta potential of particles was +22.6 ± 2 and +6.1 ± 0.1 mV for the PC/**3k** and PC/**3l** formulations, respectively. The surface charge of liposomes increased when liposomes were modified with phosphorus-containing amphiphiles **3k** and **3l**. This indicates incorporation of **3k** and **3l** into liposome membrane. Incorporation of ionic surfactants into liposomes was investigated [189]. All liposome formulations showed a low polydispersity index below 0.2. Varying the percentage of PC/amphiphiles, namely, increasing amphiphile **3k** and **3l** content from 2 to 10% (wt.), was carried out. The zeta potential of nanoparticles increased from +23 to +46 mV upon increasing the content of **3k** from 2% to 10% (wt.) for the PC/**3k** system. In the case of liposomal formulation PC/**3l**, the zeta potential increased from +6 to +8 mV. The charge of liposomal formulations depended both on the phosphorus-containing amphiphile content and on alkyl chain length. Most likely, this difference was associated with the amphiphiles penetration depth into the liposomal membrane, the level of incorporation into the double phospholipid bilayer, and the solubility of both amphiphiles [190]. There was a slight increase in the polydispersity when the model drug rhodamine B was encapsulated, but the polydispersity index did not exceed 0.2. The zeta potential of the PC/**3l** liposomal systems increased both in the presence of rhodamine B and with an increase (**3l**) content in formulation from +20.5 ± 2 to +26 ± 1 mV. The encapsulation efficiency of rhodamine was high, about 87%. EE increased to 93% with the growing amphiphile amount in the liposomal membrane. This increase may be related to stabilization of the liposomal membrane. All samples were stored in fridge (at 4 °C) in hermetically closed glass containers (to prevent from humidity during storage) for colloid stability studies upon three months. During the storage of 3 months, the characteristics of liposomes did not change, except for formulation PC/**3l** (with ratio 90/10, %*w*/*w*). It may be stated that **3k** and **3l** have a stabilizing effect on liposomes. The optimum concentration of **3k** and **3l** was observed. **3k** and **3l** nanoparticles and mixed self-assembles with PC formation were possible at high concentrations of **3k** and **3l** amphiphiles afer long storage.

UV spectra (A) and rhodamine B release profile (B) within time in the absence of nanoparticles (1) and encapsulated in liposomes (2–5) are shown in Figure 3. Rhodamine B (50%) was released from liposomes much slower (4 h) than in the case of aqueous solution (within 20 min). It should be noted that the release slowdown occurred in the case of liposome modified with amphiphil **3l**. This confirmed the improved stabilization of liposomal systems by amphiphil **3l**.

### 2.2. Biology 

#### 2.2.1. Cytotoxicity

Cytotoxicity against cancer and normal cell lines was tested for synthesized phosphonium salts. The cytotoxic activity is represented by IC_50_ values (the compound concentration that causes the death of 50% of cells in the experimental cell population) in Table 2.

The studied phosphonium salts showed high and moderate cytotoxicity against some cancer lines and moderate activity against normal lung embryo cells. The most significant results were obtained for the duodenal adenocarcinoma cell line (HuTu 80) and cell line PC3 (human prostate cancer). Compounds **3j** and **4c** showed anti-cancer activity against the HuTu 80 cell line comparable to doxorubicin activity. The **3j** and **4c** cytotoxic effect against the PC3 line was two times higher than doxorubicin. Phosphonium salts **3j** and **4c** awere the most active compounds against other human cancer lines, and the IC_50_ concentration values were in the range 1.1–3.7 μM. 

Selectivity index (SI) was calculated to assess the cytotoxic effect as the ratio between the IC_50_ value for normal cells and the IC_50_ value for cancer cells (Table 2). All compounds showed the highest selectivity against HuTu 80 line. SI is 4–12.3. It should be noted that **4c** has the highest SI equal to 12.3. Doxorubicin is significantly inferior to the leading compounds in selectivity.

The cytotoxic effect of liposome formulations **PC/3k** and **PC/3l** (with ratio 98/2, %*w*/*w*) in cancer cells and normal human cells was lower than for individual compounds **3k** and **3l**. 

#### 2.2.2. Induction of Apoptotic Effects by Lead Compounds

Apoptosis is one of the most significant mechanisms used for screening new anti-cancer agents. The apoptosis-inducing effect was carried out on lead compounds **3a**, **b**, **k**, **j** and **4c** showing high selectivity against HuTu-80 cells. The studies were carried out by flow cytometry at concentrations IC_50_/2 and IC_50_ (Figure 4). HuTu-80 cells actively induce apoptosis after 48-h incubation in the presence of **3a**, **b**, **k**, **j** and **4c** at concentrations IC_50_/2. The percentage of apoptotic cells in early and late phases was slightly changed with increase compound concentration up to IC_50_.

Apoptosis induction by **3a**, **b**, **k**, **j** and **4c** through the mitochondrial pathway and HuTu-80 cell line was studied by flow cytometry using the fluorescent dye JC-10 from the Mitochondria Membrane Potential Kit. JC-10 accumulated in the mitochondrial matrix and formed aggregates (J-aggregate) with red fluorescence in normal cells with a high mitochondrial membrane potential. The membrane potential decreased in apoptotic cells. JC-10 started to diffuse out from the mitochondria and converts to the monomeric form (J-monomer) and emits green fluorescence. This is recorded with a flow cytometer. A decrease the mitochondrial membrane potential of HuTu-80 cells occurs after treatment with **3a**, **b**, **k**, **j** and **4c**. The process became more pronounced with an increase of **3a**, **b**, **3k**, **j** and **4c** concentrations up to IC_50_ (Figure 5 and Figure 6). The obtained results suggest that the mechanism of **3a**, **b**, **k**, **j** and **4c** cytotoxic action is due to apoptosis induction through the internal mitochondrial pathway.

Increasing the production of reactive oxygen species (ROS) by compounds characterizes the mitochondrial apoptotic pathway. Mitochondria are a potential source and target of ROS. ROS lead to disruption of mitochondrial functions and, as a consequence, to irreversible cell damage. In this regard, the effect of test compounds in HuTu-80 cells on ROS production, using flow cytometry analysis and CellROX^®^ Deep Red flow cytometry kit was investigated. Data presented in Figure 7 show a significant increase in CellROX^®^ Deep Red fluorescence intensity. This indicates an increase in ROS production in the presence of the tested compounds. It should be noted that **3k**, **j** and **4c** were the most active to generate ROS in HuTu-80 cells as compared to **3a** and **3b** and control (untreated cells).

## 3. Materials and Methods

### 3.1. General 

The NMR spectra were recorded at 25 °C using a Bruker Avance-400 NMR spectrometer (400.0 MHz, ^1^H; 100.6 MHz, ^13^C; 162 MHz, ^31^P), a Bruker Avance-500 NMR spectrometer (500.0 MHz, ^1^H; 125.75 MHz, ^13^C; 201.7 MHz, ^31^P) and a Bruker Avance-600 NMR spectrometer (600.0 MHz, ^1^H; 150.9 MHz, ^13^C; 242 MHz, ^31^P). Chemical shifts were referenced to the residual solvent peak and reported in ppm (δ scale) and all coupling constant (*J*) values were given in Hz. IR spectra were recorded using a Bruker Vector 22 spectrometer for samples in KBr pellets or in film. MALDI mass spectra were acquired on a Bruker MALDI-MS Ultraflex III spectrometer. 2,5-Dihydroxybenzoic acid (5 mg/mL in methanol) and 4-nitroanilin (5 mg/mL in methanol) were used as a matrix. Melting points were determined on a Melting Point Apparatus Stuart SMP10. Optical rotations were determined on a Perkin Elmer 341 polarimeter (concentration c is given as g/mL). Elemental analysis was accomplished with an automated EuroVector EA3000 CHNS-O elemental analyzer (Euro-Vector, Pavia, Italy). 

The X-ray diffraction data for the crystals (**3****b**, **3c**, **3d**, **3f**–**3j**, **4a**) were collected on a Bruker D8 QUEST CCD diffractometerat temperature 100(2) K, (**3a**) and (**3e**) were collected on a Bruker Kappa Apex II CCD diffractometerat room temperature (296(2) K), in the ω and φ-scan modes using graphite monochromated MoK_α_ (λ = 0.71073 Å) radiation.

Data for the crystals (**3**–**4a**) were corrected for the absorption effect using SADABS program [191]. Data collection: images were indexed and integrated using the APEX2 data reduction package [192]. The structures were solved by direct method using SHELXT [193] and refined by the full matrix least-squares using SHELXL programs [194].

The X-ray diffraction data for the crystals (**3****e**, ***S***) and (**3e**, ***R***) were collected on a Enraf–Nonius CAD-4 diffractometer at room temperature (296(2) K), in the ω/2θ-scan modes using graphite monochromated MoK_α_ (λ = 0.71073 Å) radiation. The structure was solved by direct method using and refined by the full matrix least-squares using MolEN [195] programs. Then the structure wasrefined by the full matrix least-squares using SHELXL programs [194] in the WinGX program package [196].

The X-ray diffraction data for the crystal (**7**) were collected on a XtaLAB Synergy, Single source at home/near, HyPix diffractometerat temperature 100(2) Kusing graphite monochromated CuK_α_ (λ = 1.54184 Å) radiation). The structure was solved by the direct method using SHELXS and refined by the full matrix least-squares using SHELXL programs. Hydrogen atoms in all structures were inserted at calculated positions and refined as riding atoms.Hydrogen atoms in the hydroxyl-groups were solved from difference Fourier maps and refined with fixed bond length and angles with rotation around C–O bonds. Analysis of the intermolecular interactions was performed using the program PLATON [197]. Mercury program package [198] was used for figures preparation.

Crystallographic data for the investigated structuresare seen in the Appendix A and have been deposited in the Cambridge Crystallographic Data Centre as supplementary publication nos. CCDC 2110389-2110402, respectively. Copies of the data can be obtained free of charge upon application to the CCDC (12 Union Road, Cambridge CB2 1EZ UK. Fax: (internat.) +44-1223/336-033; E-mail: deposit@ccdc.cam.ac.uk).

The synthesis of compounds **3d**, **7** and their purity were monitored by TLC on Sorbfil plates (IMID Ltd., Krasnodar, Russian). The TLC plates were visualized by UV. The targeted compounds **3d**, **7** were isolated using column chromatography on silica gel (60A, 60–200 μm, Acros, Belgium). Solvents were purified and dried by standard protocols.

### 3.2. Chemistry

#### 3.2.1. General Procedure for the Synthesis of 1-Chloro-3-alkoxypropan-2-ol (**5a****–d**) 

Compounds **5a****–****d** were synthesized according to a modified procedure [199]. To the corresponding alcohol (0.4 mol) was added 5 wt % of boron trifluoride etherate and epichlorohydrin (0.1 mol) and the reaction mixture was refluxed for 15 h until the reaction was completed. The mixture was then cooled at room temperature and diluted with water. The solution was exhaustively extracted with dichloromethane. The combined extracts were washed with brine and dried with anhydrous sodium sulfate, andfinally, the dichloromethane was evaporated in vacuo. 1-Chloro-3-alkoxypropan-2-ol was purified by vacuum distillation.

**1-Chloro-3-methoxypropan-2-ol** (**5a**). Yield: 67%; bp 87–90 °C (30 mmHg); *n*_D_^20^ 1.4424; ^1^H NMR spectrum (600 MHz, CDCl_3_) δ ppm (*J* Hz) 3.97 (m, H^2^, 1H, ^3^*J*_HH_ 5.2–5.7), 3.58 and 3.62 (two m, H^3^, 2H, *AB*-part of *ABX*-spectrum, ^2^*J*_H_^3A^_H_^3B^ 11.1, ^3^*J*_H_^2^_H_^3^ 5.7), 3.50 (m, H^1^, *AB*-part of *ABX*-spectrum, ^2^*J*_H_^1A^_H_^1B^ 10.4), 3.40 (s, H^1′^, 3H), 2.65 br. s (OH, 1H). Found: anal. C, 38.43; H, 7.11; Cl, 28.30%, calcd. for C_4_H_9_ClO_2_, C, 38.57; H, 7.28; Cl, 28.46%.

**1-Chloro-3-ethoxypropan-2-ol** (**5b**). Yield: 85%; bp 93–97 °C (25 mmHg); *n*_D_^20^ 1.4372; ^1^H NMR spectrum (600 MHz, CDCl_3_) δ ppm (*J* Hz) 3.97 (m, H^2^, 1H, ^3^*J*_HH_ 5.0–5.6), 3.60 and 3.64 (two m, H^3^, 2H, *AB*-part of *ABX*-spectrum, ^2^*J*_H_^3A^_H_^3B^ 11.1, ^3^*J*_H_^2^_H_^3^ 5.6), 3.54–3.56 (m, H^1^ and H^1′^, 4H), 2.53 (br. s, OH, 1H), 1.22 (t, H^2′^, 3H, ^3^*J*_HH_ 7.0). Found: anal. C, 43.12; H, 7.88; Cl, 25.57%, calcd. for C_5_H_11_ClO_2_, C, 43.33; H, 8.00; Cl, 25.58%.

**1-Chloro-3-(decyloxy)propan-2-ol** (**5****d**). Yield: 72%; bp 128–130 (0.1 mmHg); *n*_D_^20^ 1.4530; IRS (film, cm^−1^): 3418, 2954, 2925, 2855, 1598, 1465, 1430, 1378, 1299, 1260, 1120, 1074, 945, 910, 843, 753, 722, 624, 572; ^1^H NMR spectrum (600 MHz, CDCl_3_) δ ppm (*J* Hz) 3.97 (m, H^2^, 1H, ^3^*J*_HH_ 5.4, ^3^*J*_HH_ 5.4), 3.60 and 3.64 (two m, H^3^, 2H, *AB*-part of *ABX*-spectrum, ^2^*J*_H_^A^_H_^B^ 11.0, ^3^*J*_H_^2^_H_^A^ = ^3^*J*_H_^2^_H_^B^ 5.4), 3.53 (m, H^1′^, 2H, *AB*-part of *ABX*_2_-spectrum), 3.48 (m, H^1^, 2H, *AB*-part of *ABX*-spectrum), 2.52 (br. s, OH, 1H), 1.28–1.31 (m, H^3′^-H^9′^, 14H), 0.89 (t, H^10′^, 3H, ^3^*J*_HH_ 7.1); ^13^C NMR spectrum (100.6 MHz, CDCl_3_) δ_C_ ppm (*J* Hz) 71.75 (tm (s) (here and below, the form of the signal in NMR^13^C-{^1^H} spectra is indicated in parentheses), C^3^, ^1^*J*_HC_ 140.4), 71.37 (tm (s), C^1′^, ^1^*J*_HC_ 141.7, ^2^*J*_HCC_ 2.9, ^3^*J*_HCCC_ 2.8, ^3^*J*_HCCC_ 2.8), 760.26 (dtt (s), C^2^, ^1^*J*_HC_ 145.6, ^2^*J*_HCC_ 2.6–2.8, ^2^*J*_HCC_ 2.6–2.8), 45.97 (brtt (s), C^1^, ^1^*J*_HC_ 150.8, ^3^*J*_HC_^3^_CC_ 3.6, ^2^*J*_HCC_ 0), 31.92 (tm (s), C^2′^, ^1^*J*_HC_ 125.7, ^3^*J*_HCCC_ 4.2–5.0, ^2^*J*_HCC_ 2.8–3.2), 29.63 (tm (s), C^3′^, ^1^*J*_HC_ 124.0–125.0), 29.60 (tm (s), C^4′^, ^1^*J*_HC_ 124.0–125.0), 29.55 (tm (s), C^5′^, ^1^*J*_HC_ 124.0–125.0), 29.48 (tm (s), C^6′^, ^1^*J*_HC_ 124.0–125.0), 29.35 (tm (s), C^7′^, ^1^*J*_HC_ 125.1), 26.06 (tm (s), C^8′^, ^1^*J*_HC_ 125.1), 22.70 (tm (s), C^9′^, ^1^*J*_HC_ 125.0, ^3^*J*_HCCC_ 3.8–4.0, ^2^*J*_HCC_ 2.6–2.8), 14.11 (qm (s), C^10′^, ^1^*J*_HC_ 124.5). Found: anal. C, 62.14; H, 10.82; Cl, 14.09%, calcd. for C_13_H_27_ClO_2_, C, 62.26; H, 10.85; Cl, 14.13%.

#### 3.2.2. General Procedure for the Synthesis of 2-(Alkoxymethyl)oxirane (**2f****–****j**)

Compounds **2f****–****j** were synthesized according to a modified procedure [180]. Sodium hydroxide (0.75 mmol) was added portion to a stirred solution of corresponding 3-alkoxy-1-chloropropan-2-ol(0.5 mol) in ether (500 mL) at 0 °C. The reaction mixture was stirred for 8 h at room temperature a then poured onto iced water (250 mL) and the layers formed were separated. The aqueous layer was extracted with ether, the combined ether solutions were evaporated in vacuum; 2-(alkoxymethyl)oxirane was purified by vacuum distillation.

**3-(Methoxymethyl)oxirane**(**2f**). Yield: 88%; bp 58–60 °C (90 mmHg); *n*_D_^20^ 1.4062; IRS (film, cm^−1^): 3502, 3057, 2993, 2930, 2885, 2823, 1454, 1385, 1345, 1256, 1200, 1163, 1136, 1108, 1058, 988, 963, 939, 904, 848, 803, 764; ^1^H NMR spectrum (400 MHz, CDCl_3_) δ ppm (*J* Hz) 3.64 (dd, H^3M^, 1H, *M*-part of *AMX*-spectrum, ^2^*J*_H_^3A^_H_^3M^ 11.4, ^3^*J*_H_^2^_H_^3M^ 3.0), 3.28 (dd, H^3A^, 1H, *A*-part of *AMX*-spectrum, ^2^*J*_H_^3M^_H_^3A^ 11.1, ^3^*J*_H_^2^_H_^3A^ 5.9), 3.36 (s, H^1′^, 3H), 3.09 (m, H^2^, ^3^*J*_H_^3A^_H_^2^ 5.9, ^3^*J*_H_^1M^_H_^2^ 4.2, ^3^*J*_H_^3M^_H_^2^ 3.0, ^3^*J*_H_^1A^_H_^2^ 2.7), 2.74 (dd, H^1M^, 1H, ^2^*J*_H_^1A^_H_^1M^ 5.1, ^3^*J*_H_^2^_H_^1M^ 4.2), 2.56 (dd, H^1A^, 1H, ^2^*J*_H_^1M^_H_^1A^ 5.1, ^3^*J*_H_^2^_H_^1A^ 2.7). Found: anal. C, 54.48; H, 9.10%, calcd. for C_4_H_8_O_2_, C, 54.53; H, 9.15%.

**3-(Ethoxymethyl)oxirane** (**2g**). Yield: 90%; bp 55–60 °C (60 mmHg), *n*_D_^20^ 1.4110; IRS (film, cm^−1^): 3585, 3517, 3055, 2978, 2931, 2872, 1486, 1446, 1396, 1335, 1254, 1161, 1113, 1019, 992, 915, 880, 856, 800, 762; ^1^H NMR spectrum (400 MHz, CDCl_3_) δ ppm (*J* Hz) 3.64 (dd, H^3M^, 1H, *M*-part of *AMX*-spectrum, ^2^*J*_H_^3A^_H_^3M^ 11.4, ^3^*J*_H_^2^_H_^3M^ 3.0), 3.32 (dd, H^3A^, 1H, *A*-part of *AMX*-spectrum, ^2^*J*_H_^3M^_H_^3A^ 11.4, ^3^*J*_H_^2^_H_^3A^ 5.8), 3.48 and 3.51 (two m, H^1^ spectrum (400 MHz, CDCl_3_) δ ppm, 2H, *AB*-part of *ABX*_3_-spectrum, ^2^*J*_H_^1′A^_H_^1′B^ 9.4, ^3^*J*_HH_ 7.0), 3.08 (m, H^2^, 1H, ^3^*J*_H_^3A^_H_^2^ 5.8, ^3^*J*_H_^1M^_H_^2^ 4.2, ^3^*J*_H_^3M^_H_^2^ 3.0, ^3^*J*_H_^1A^_H_^2^ 2.8), 2.72 (dd, H^1M^, 1H, ^2^*J*_H_^1A^_H_^1M^ 5.1, ^3^*J*_H_^2^_H_^1M^ 4.2), 2.54 (dd, H^1A^, 1H, ^2^*J*_H_^1M^_H_^1A^ 5.1, ^3^*J*_H_^2^_H_^1A^ 2.7), 1.16 (t, H^2′^, 3H, ^3^*J*_HH_ 7.0). Found: anal. C, 58.72; H, 9.83%, calcd. for C_5_H_10_O_2_, C, 58.80; H, 9.87%.

**3-((****2′,2′,3′,3′-Tetrafluoropropanoxy)methyl)oxirane** (**2h**) obtained according to the method [200,201]. Yield: 80%; bp 100–110 °C (50 mmHg), *n*_D_^20^ 1.3650; IRS (film, cm^−1^): 3501, 3067, 3069, 2933, 1466, 1445, 1405, 1344, 1279, 1231, 1266, 1109, 998, 939, 903, 833, 764, 746, 690; ^1^H NMR spectrum (400 MHz, CDCl_3_) δ ppm (*J* Hz) 5.92 (tt, H^3′^, 1H, ^2^*J*_F_^3′^_H_^3′^ 53.2, ^3^*J*_F_^2′^_H_^3′^ 5.0), 3.86 and 3.92 (two br m, H^1′^, 2H, *AB*-part of *ABX*_2_-spectrum, ^2^*J*_H_^1′A^_H_^1′B^ 11.5, ^3^*J*_F_^2′^_H_^1′^ 12.0), 3.90 (dd, H^3M^, 1H, ^2^*J*_H_^3A^_H_^3M^ 11.8, ^3^*J*_H_^2^_H_^3M^ 2.4), 3.45 (dd, H^3A^, 1H, ^2^*J*_H_^3M^_H_^3A^ 11.8, ^3^*J*_H_^2^_H_^3A^ 6.1), 3.13 (m, H^2^, 1H), 2.79 (dd, H^1M^, 1H, ^2^*J*_H_^1A^_H_^1M^ 4.9, ^3^*J*_H_^2^_H_^1M^ 4.2), 2.58 (dd, H^1A^, 1H, ^2^*J*_H1MH1A_ 4.9, ^3^*J*_H2H1A_ 2.6). Found: anal. C, 38.43; H, 4.27%, calcd. for C_6_H_8_F_4_O_2_, C, 38.31; H, 4.29%.

**3-((Allyloxy)methyl)oxirane** (**2i**). Yield: 82%; bp 68–69 °C (20 mmHg), *n*_D_^20^ 1.4369; IRS (film, cm^−1^): 3058, 2999, 2924, 2858, 1647, 1457, 1421, 1338, 1253, 1098, 994, 924, 855, 802, 763; ^1^H NMR spectrum (400 MHz, CDCl_3_) δ ppm (*J* Hz) 5.88 (ddt, H^2′^, 1H, ^3^*J*_H_^3′^_(X)H_ 17.2, ^3^*J*_H_^3′A^_H_ 10.4, ^3^*J*_H_^1′^_H_ 5.6), 5.26 (ddt, H^3′X^, 1H, ^3^*J*_H_^2′^_H_ 17.2, ^2^*J*_HH_ 1.7, ^4^*J*_H_^1′^_H_ 1.6), 5.16 (ddt, H^3′A^, 1H, ^3^*J*_H_^2′^_H_ 10.4, ^2^*J*_HH_ 1.6, ^4^*J*_H_^1′^_H_ 1.2), 4.04 (dddd, H^1′A^, 1H, ^2^*J* 12.8, ^3^*J*_H_^2′^_H_ 5.6, ^4^*J*_H_^3′^_H_ 1.4, ^4^*J*_H_^3′^_H_ 1.3), 3.99 (dddd, H^1′B^, 1H, ^2^*J* 12.8, ^3^*J*_H_^2′^_H_ 5.6, ^4^*J*_H_^3′^_H_ 1.4, ^4^*J*_H_^3′^_H_ 1.3), 3.64 (dd, H^3M^, 1H, *M*-part of *AMX*-spectrum, ^2^*J*_H_^3A^_H_^3M^ 11.4, ^3^*J*_H_^2^_H_^3M^ 3.1), 3.38 (dd, H^3A^, 1H, *A*-part of *AMX*-spectrum, ^2^*J*_H_^3M^_H_^3A^ 11.4, ^3^*J*_H_^2^_H_^3A^ 5.8), 3.12 (m, H^2^, 1H, ^3^*J*_H_^3A^_H_^2^ 5.8, ^3^*J*_H_^1M^_H_^2^ 4.1, ^3^*J*_H_^3M^_H_^2^ 3.0, ^3^*J*_H_^1A^_H_^2^ 2.7), 2.76 (dd, H^1M^, 1H, ^2^*J*_H_^1A^_H_^1M^ 5.0, ^3^*J*_H_^2^_H_^1M^ 4.2), 2.58 (dd, H^1A^, 1H, ^2^*J*_H_^1M^_H_^1A^ 5.0, ^3^*J*_H_^2^_H_^1A^ 2.7). Found: anal. C, 63.02; H, 8.35%, calcd. for C_6_H_10_O_2_, C, 63.14; H, 8.83%.

**3-((Decyloxy)methyl)oxirane** (**2j**). Yield: 90%; bp 87–92 °C (0.1 mmHg), *n*_D_^20^ 1.4430; IRS (film, cm^−1^): 3051, 2926, 2855, 1613, 1467, 1338, 1253, 1158, 1111, 913, 848, 762, 722; ^1^H NMR spectrum (400 MHz, CDCl_3_) δ ppm (*J* Hz) 3.47 (ddd, H^3B^, 1H, *B*-part of *ABX*-spectrum, ^2^*J*_H_^3A^_H_^3B^ 11.5, ^3^*J*_H_^2^_H_^3B^ 3.0, ^4^*J*_H_^1^_H_^3B^ 1.4), 3.13 (ddd, H^3A^, 1H, *A*-part of *ABX*-spectrum, ^2^*J*_H_^3B^_H_^3A^ 11.5, ^3^*J*_H_^2^_H_^3A^ 5.8, ^4^*J*_H_^1^_H_^3A^ 1.4), 3.23–3.26 (m, H^1′^, 2H, *AB*-part of *ABX*_2_-spectrum, ^2^*J*_H_^1′A^_H_^1′B^ 10.0–11.0, ^3^*J*_H_^2′^_H_^1′^ 6.7, ^4^*J*_HH_^1′^ 1.3), 2.90 (m, H^2^, 1H), 2.55 (m, H^1B^, 1H, ^2^*J*_H_^1A^_H_^1B^ 5.2, ^3^*J*_H_^2^_H_^1B^ 4.1, ^4^*J*_H_^3B^_H_^1B^ 1.4), 2.36 (m, H^1A^, 1H, ^2^*J*_H_^1B^_H_^1A^ 5.2, ^3^*J*_H_^2^_H_^1A^ 2.6), 1.35 (m, H^2′^, 2H, ^3^*J*_HH_ 6.7), 1.04–1.10 (m, H^3′^-H^9′^, 14H), 0.65 (t, H^10′^, 3H, ^3^*J*_HH_ 6.7); ^13^C-{^1^H} NMR spectrum (100.6 MHz, CDCl_3_) δ_C_ ppm (*J* Hz) 71.58 (s, C^3^), 71.40 (s, C^1′^), 50.75 (s, C^2^), 44.06 (s, C^1^), 31.86 (s, C^2′^), 29.66 (s, C^3′^), 29.56 (s, C^4′^), 29.53 (s, C^5′^), 29.44 (s, C^6′^), 29.29 (s, C^7′^), 26.05 (s, C^8′^), 22.62 (s, C^9′^), 14.01 (s, C^10′^). Found: anal. C, 72.70; H, 12.26%, calcd. for C_13_H_26_O_2_, C, 72.85; H, 12.23%.

#### 3.2.3. General Procedure for the Synthesis of 3-Alkoxy-1-iodopropan-2-ols (**6a**–**d**) 

Compounds **6a**–**d** were synthesized according to a modified procedure [202]. To a solution of the corresponding previously obtained compound **5a**–**d** (3 mmol) in CH_3_CN (5 mL) was added NaI (12 mmol) and 10% 18-crown-6. The reaction mixture was stirredunder reflux for 20 h. After cooling the mixture to room temperature, the precipitate was filtered. The mixture poured onto water (40 mL) and was extracted three times with ether. The combined ether solutions were washed with H_2_O and then organic layer dried over MgSO_4_, filtered, and concentrated in vacuo; 3-alkoxy-1-iodopropan-2-olwas purified by vacuum distillation.

**1-Iodo-3-methoxypropan-2-ol** (**6a**). Yield: 87%; bp 115–117 °C (30 mmHg); *n*_D_^20^ 1.5227; IRS (film, cm^−1^): 3417 very br. s. (OH), 2984, 2927, 2892, 2825, 1638, 1453, 1415, 1378, 1329, 1285, 1251, 1198, 1185, 1119, 1065, 1031, 1006, 964, 933, 890, 861, 807, 745, 626, 542, 515, 465. ^1^H NMR spectrum (400 MHz, CDCl_3_) δ ppm (*J* Hz) 3.75 (m, H^2^, 1H, ^3^*J*_HH_ 5.6, ^3^*J*_HH_ 5.6), 3.46 and 3.48 (two m, H^3^, 2H, *AB*-part of *ABX*-spectrum, ^2^*J*_H3AH3B_ 10.6, ^3^*J*_H2H3_ 5.6), 3.38 (s, H^1′^, 3H), 3.25 and 3.29 (two m, H^1^, 2H, *AB*-part of *ABX*-spectrum, ^2^*J*_H1AH1B_ 10.2, ^3^*J*_H2H1_ 5.6), 2.94 (very br s, OH, 1H); anal. C, 22.04; H, 4.14; I 58.30%. Found: calcd. for C_4_H_9_IO_2_, C, 22.24; H, 4.20; I 58.75%.

**1-Ethoxy-3-iodopropan-2-ol** (**6b**). Yield: 92%; bp 122–125 °C (22 mmHg); *n*_D_^20^ 1.5180; IRS (film, cm^−1^): 3417 very br. s. (OH), 2974, 2928, 2871, 1630, 1484, 1444, 1413, 1382, 1354, 1328, 1276, 1237, 1186, 1117, 1075, 1034, 1007, 936, 906, 867, 807, 627, 596, 514, 448. ^1^H NMR spectrum (400 MHz, CDCl_3_) δ ppm (*J* Hz) 3.74 (tdd, H^2^, 1H, ^3^*J*_H_^3A^_H_^2^ 5.7, ^3^*J*_H_^3B^_H_^2^ 4.6, ^3^*J*_H_^1^_H_^2^ 5.7), 3.52 and 3.53 (two m, H^1′^, 2H, *AB*-part of *ABX*_3_-spectrum, ^2^*J*_H_^1′A^_H_^1′B^ 10.0, ^3^*J*_HH_ 7.0), 3.50 and 3.51 (two m, H^3^, 2H, *AB*-part of *ABX*-spectrum, ^2^*J*_H_^3A^_H_^3B^ 9.7, ^3^*J*_H_^2^_H_^3A^ 5.7, ^3^*J*_H_^2^_H_^3B^ 4.6), 3.25 and 3.31 (two m, H^1^, 2H, *AB*-part of *ABX*-spectrum, ^2^*J*_H_^1A^_H_^1B^ 10.2, ^3^*J*_H_^2^_H_^1A^ 5.7, ^3^*J*_H_^2^_H_^1B^ 5.7), 1.19 (t, H^2′^, 3H, ^3^*J*_HH_ 7.0). Found: anal. C, 25.97; H, 4.80; I 55.00%, calcd. for C_5_H_11_IO_2_, C, 26.11; H, 4.82; I 55.16%.

**1-(Allyloxy)-3-iodopropan-2-ol** (**6c**). Yield: 86%; *n*_D_^20^ 1.5305; bp 75–76 °C (0.1 mmHg); IRS (film, cm^−1^): 3418, 3079, 3013, 2903, 2860, 1646, 1470, 1450, 1420, 1350, 1329, 1264, 1187, 1108, 1007, 931, 879, 807; ^1^H NMR spectrum (400 MHz, CDCl_3_) δ ppm (*J* Hz) 5.91 (ddt, H^2′^, 1H, ^3^*J*_H_^3′X^_H_ 17.2, ^3^*J*_H_^3′A^_H_ 10.4, ^3^*J*_H_^1′^_H_ 5.7), 5.29 (ddt, H^3′X^, 1H, ^3^*J*_H_^2′^_H_ 17.2, ^2^*J*_HH_ 1.7, ^4^*J*_H_^1′^_H_ 1.6), 5.22 (ddt, H^3′A^, 1H, ^3^*J*_H_^2′^_H_ 10.4, ^2^*J*_HH_ 1.6, ^4^*J*_H_^1′^_H_ 1.4), 4.04 (ddd, H^1′^, 2H, ^3^*J*_H_^2′^_H_ 5.7, ^4^*J*_H_^3′^_H_ 1.6, ^4^*J*_H_^3′^_H_ 1.4), 3.79 (tdd, H^2^, 1H, ^3^*J*_H_^3^_H_ 5.8, ^3^*J*_H_^3^_H_ 5.6, ^3^*J*_H_^1^_H_ 5.6), 3.55 (m, H^1^, 2H, *AB*-part of *ABX*-spectrum, ^3^*J*_HH_ 5.6), 3.28 and 3.34 (two m, H^3^, 2H, *AB*-part of *ABX*-spectrum, ^2^*J*_H3AH3B_ 10.2, ^3^*J*_H2H1A_ 5.8, ^3^*J*_H2H1B_ 5.6), 2.64 br. s (OH, 1H). Found: anal. C, 29.63; H, 4.12; I 52.37%, calcd. for C_6_H_11_IO_2_, C, 29.77; H, 4.58; I 52.43%.

**1-(Decyloxy)-3-iodopropan-2-ol** (**6****d**). Yield: 82%; *n*_D_^20^ 1.4930; bp 132–134 °C (0.18 mmHg); IRS (film, cm^−1^): 3417 very br. s. (OH), 2953, 2923, 2854, 1725, 1640, 1464, 1415, 1377, 1328, 1259, 1239, 1185, 1118, 1034, 1007, 939, 877, 808, 721, 629, 554, 537, 453. ^1^H NMR spectrum (400 MHz, CDCl_3_) δ ppm (*J* Hz) 3.77 (tt,H^2^, 1H, ^3^*J*_HH_ 5.6, ^3^*J*_HH_ 5.6), 350 and 3.53 (two m, H^3^, 2H, *AB*-part of *ABX*-spectrum, ^2^*J*_H_^3A^_H_^3B^ 10.5, ^3^*J*_H_^2^_H_^3^ 5.6), 3.49 (td, H^2′^, 2H, ^3^*J*_HH_ 6.6–6.7, ^4^*J*_HH_ 1.4), 3.28 and 3.34 (two m, H^1^, 2H, *AB*-part of *ABX*-spectrum, ^2^*J*_H_^1A^_H_^1B^ 10.1, ^3^*J*_H_^2^_H_^1^ 5.6), 2.55 (br s, OH, 1H), 1.59 (m, H^3′^, 2H, ^3^*J*_HH_ 6.7–7.1), 1.28–1.32 (m, H^3′^-H^9′^, 14H), 0.90 (t, H^10′^, 3H, ^3^*J*_HH_ 7.1). Found: anal. C, 45.60; H, 7.83; I 37.02%, calcd. for C_13_H_27_IO_2_, C, 45.62; H, 7.95; I 37.08%.

#### 3.2.4. General Procedure for the Synthesis of Triphenylphosphonium Salts (**3a**–**e**, **3f**–**j**)

To a solution of triphenylphosphine (1 mmol) in anhydrous CH_2_Cl_2_ (20 mL), under argon, CF_3_SO_3_H (1 mmol) was added at −10 ÷ 0 °C. After 10 min at −10 ÷ 0 °C, a solution of **2a–j** (1 mmol) in anhydrous CH_2_Cl_2_ (20 mL) was added dropwise. The solution was stirred under argon atmosphere at room temperature for 15 min and then a solution concentrated in vacuo. Petroleum ether (5 mL) was added to the reaction mixture. The resulting precipitate was washed with diethyl ether (3 mL) and dried in vacuo to give the pure TPP salt.

**(3-Chloro-2-hydroxypropyl)triphenylphosphonium trifluoromethanesulfonate** (**3a**). Yield: 92%; mp 146–147 °C; IRS (cm^−1^): 3365–3370 very br s (ν_OH_), 1585, 1482, 1480, 1340, 1315, 1285, 1260, 1255, 1225, 1199, 1180, 1176, 1113, 1085, 1030, 1002, 928, 841, 790, 755, 750, 725, 713, 680, 645, 575, 535, 520, 510, 498, 466; ^1^H NMR spectrum (400 MHz, CDCl_3_) δ ppm (*J* Hz) 7.78 (m, H*^p^*, 3H, ^3^*J*_HH_ 7.4, ^5^*J*_HH_ 2.0), 7.72 (m, H*^o^*, 6H, ^3^*J*_PH_ 12.8, ^3^*J*_HH_ 7.6), 7.68 (m, H*^m^*, 6H, ^3^*J*_HH_ 7.4–7.6, ^4^*J*_PH_ 3.7), 4.81 (very br. s, OH, 1H), 4.17 (m, H^2^, 1H, ^3^*J*_H_^1B^_H_^2^ 10.7, ^3^*J*_H_^3^_H_^2^ 5.4, ^2^*J*_H_^1A^_H_^2^ 2.5), 3.70–3.71 (m, H^3^, 2H), 3.67 (ddd, H^1B^, ^2^*J*_H_^1A^_H_^1B^ 15.6, ^2^*J*_PH_^1B^ 11.7, ^3^*J*_H_^2^_H_^1B^ 10.7), 3.48 (ddd, H^1A^, 1H, ^2^*J*_H_^1B^_H_^1A^ 15.6, ^2^*J*_PH_^1A^ 13.4, ^2^*J*_H_^2^_H_^1A^ 2.5); ^13^C NMR spectrum (100.6 MHz, acetone-*d*_6_ + 10% CDCl_3_) δ_C_ ppm (*J* Hz) 135.50 (dtd (d), C*^p^*, ^1^*J*_HC_ 164.3, ^3^*J*_HCCC_ 7.2, ^4^*J*_PCCCC_ 2.9), 134.70 (dddd (d), C*^o^*, ^1^*J*_HC_ 166.3, ^2^*J*_PCC_ 10.3, ^3^*J*_HCCC_ 7.6, ^3^*J*_HCCC_ 7.0), 130.67 (ddd (d), C*^m^*, ^1^*J*_HC_ 166.8, ^3^*J*_PCCC_ 12.8, ^3^*J*_HCCC_ 7.5), 121.86 (q (q), CF_3_, ^1^*J*_FC_ 321.2), 120.08 (dt (d), C*^i^*, ^1^*J*_PC_ 87.6, ^3^*J*_HCCC_ 8.3), 66.92 (dm (d), C^2^, ^1^*J*_HC_ 147.3, ^2^*J*_PCC_ 5.4, ^2^*J*_HCC_ 2.8–3.0, ^2^*J*_HCC_ 2.8–3.0), 28.42 (tdm (d), C^1^, ^1^*J*_HC_ 131.3, ^1^*J*_PC_ 55.6), 49.89 (tddt (d), C^3^, ^1^*J*_HC_ 152.0, ^3^*J*_PCCC_ 17.7, ^2^*J*_HCC_ 3.5, ^3^*J*_HCCC_ 2.0–2.2); ^19^F NMR spectrum (376.5 MHz, CDCl_3_) δ_F_ −78.26 ppm (s); ^31^P-{^1^H} NMR spectrum (CH_2_Cl_2_) δ_P_ 22.5 ppm; MALDI-MS, *m*/*z* 355.5 [M − CF_3_SO_3_]^+^. Calcd. for C_21_H_21_ClO_2_P^+^: 355.8. Found: anal. C 52.55; H 4.16; Cl 6.98%, calcd. for C_22_H_21_ClF_3_O_4_PS, C 52.34; H 4.19, Cl 7.02%.

***R*****-(3-Chloro-2-hydroxypropyl)triphenylphosphonium trifluoromethanesulfonate** (**3e**, ***R***). Yield: 93%; mp 137-138 °C; [α]_D_^20^ +26.0 (*c* 1.0, CHCl_3_); ^1^H NMR spectrum (400 MHz, CDCl_3_) δ ppm (*J* Hz) 7.79 m (H*^p^*, 3H, ^3^*J*_HH_ 7.3, ^5^*J*_HH_ 1.9), 7.72 m (H*^o^*, 6H, ^3^*J*_PH_ 12.7, ^3^*J*_HH_ 7.6), 7.69 m (H*^m^*, 6H, ^3^*J*_HH_ 7.3–7.6, ^4^*J*_PH_ 3.7), 4.37 br. s (OH, 1H), 4.18 br. m (H^2^, 1H), 3.67–3.70 m (H^1B^, H^3^, 3H, ^2^*J*_H_^1A^_H_^1B^ 15.5, ^2^*J*_PH_^1B^ 11.3, ^3^*J*_H_^2^_H_^1B^ 10.3), 3.46 ddd (H^1A^, 1H, ^2^*J*_H_^1B^_H_^1A^ 15.6, ^2^*J*_PH_^1B^ 13.2, ^2^*J*_H_^2^_H_^1B^ 2.2).^31^P-{^1^H} NMR spectrum (162.0 MHz, CDCl_3_): δ_P_ 24.2 ppm.

***S*****-(3-Chloro-2-hydroxypropyl)triphenylphosphonium trifluoromethanesulfonate** (**3e**, ***S***). Yield: 91%; mp 138 °C; [α]_D_^20^ −22.2 (*c* 1.0, CHCl_3_); ^1^H NMR spectrum (400 MHz, CDCl_3_) δ ppm (*J* Hz) 7.81 m (H*^p^*, 3H, ^3^*J*_HH_ 7.3, ^5^*J*_HH_ 1.9), 7.72 m (H*^o^*, 6H, ^3^*J*_PH_ 12.7, ^3^*J*_HH_ 7.6), 7.69 m (H*^m^*, 6H, ^3^*J*_HH_ 7.3–7.6, ^4^*J*_PH_ 3.7), 4.19 br. m (H^2^, 1H), 3.70–3.73 m (H^1B^, H^3^, 3H, ^2^*J*_H_^1A^_H_^1B^ 15.5, ^2^*J*_PH_^1B^ 11.3, ^3^*J*_H_^2^_H_^1B^ 10.3), 3.63 br. s (OH, 1H), 3.48 ddd (H^1A^, 1H, ^2^*J*_H_^1B^_H_^1A^ 15.6, ^2^*J*_PH1A_ 13.2, ^2^*J*_H2H1A_ 2.2). ^31^P-{^1^H} NMR spectrum (162.0 MHz, CDCl_3_): δ_P_ 24.2 ppm.

**(3-Bromo-2-hydroxypropyl)triphenylphosphonium trifluoromethanesulfonate** (**3b**). Yield: 95%; mp 161–162 °C (from methanol); IRS (cm^−1^): 3358 (OH), 2961, 2914, 1588, 1485, 1439, 1419, 1400, 1335, 1313, 1283, 1247, 1222, 1170, 1164, 1110, 1068, 1026, 997, 913, 836, 789, 758, 750, 724, 716, 692, 639, 574, 550, 515, 505, 492, 464, 428, 410. ^1^H NMR spectrum (250 MHz, CDCl_3_) δppm (*J* Hz) 7.67–7.77 (m, C_6_H_5_, 15H), 4.16 (m, H^2^, 1H), 3.66–3.70 (two m, H^3^, H^1B^, ^3^*J*_H_^2^_H_ 5.4; H^1B^, ^2^*J*_H_^A^_H_^B^ 14.7, ^3^*J*_PH_^B^ 11.9, ^3^*J*_H_^2^_H_^B^ 11.0), 3.44 (ddd, H^1A^, ^2^*J*_H_^B^_H_^A^ 14.7, ^3^*J*_PH_^A^ 13.6, ^3^*J*_H_^2^_H_^A^ 2.4); ^1^H NMR spectrum (400 MHz, CDCl_3_ + 10% acetone-*d*_6_) δ ppm (*J* Hz) 7.70 and 7.84 (two m, C_6_H_5_, 15H), 4.19 (m, H^2^, 1H), 3.94 (ddd, H^1B^, 1H, *B*-part of *ABX*-spectru+m, ^3^*J*_H_^2^_H_^1B^ 10.6, ^2^*J*_H_^1A^_H_^1B^ 15.6, ^3^*J*_PH_^1B^ 12.2), 3.73 (ddd, H^1A^, 1H, *A*-part of *ABX*-spectrum, ^2^*J*_H_^1A^_H_^1B^ 15.6, ^3^*J*_PH_^1A^ 12.2, ^3^*J*_H_^2^_H_^1A^ 2.8), 3.70 (m, H^3^, 2H, *AB*-part of *ABX*-spectrum); ^1^H NMR spectrum (400 MHz, CDCl_3_) δ ppm (*J* Hz) 7.69–7.85 (m, C_6_H_5_, 15H), 4.20 (m, H^2^, 1H), 3.76–3.86 (m, H^1^, 1H), 3.70 (m, H^3^, 2H), 3.51–3.59 (m, H^1^, 1H); ^13^C NMR spectrum (100.6 MHz, CDCl_3_ + 10% acetone-*d*_6_) δ_C_ ppm (*J* Hz) 134.98 (dtd (d), C*^p^*, ^1^*J*_HC_ 163.9, ^3^*J*_HCCC_ 7.0, ^4^*J*_PCCCC_ 3.1), 133.84 (dddd (d), C^0^, ^1^*J*_HC_ 164.5, ^2^*J*_PCC_ 10.3, ^3^*J*_HCCC_ 7.6, ^3^*J*_HCCC_ 6.5), 130.25 (ddd (d), C*^m^*, ^1^*J*_HC_ 165.9, ^3^*J*_PCCC_ 12.8, ^3^*J*_HCCC_ 7.2), 120.51 (q (q), CF_3_, ^1^*J*_FC_ 319.8), 118.54 (dt (d), C*^i^*, ^1^*J*_PC_ 87.5, ^3^*J*_HCCC_ 8.0), 65.88 (dm (d), C^2^, ^1^*J*_HC_ 150.5, ^2^*J*_PCC_ 5.0), 38.05 (tdm (d), C^3^, ^1^*J*_HC_ 151.6, ^3^*J*_PCCC_ 16.6, ^2^*J*_HCC_ 2.6, ^3^*J*_HCCC_ 2.4–2.6), 29.41 (tdm (d), C^1^, ^1^*J*_HC_ 132.0, ^1^*J*_PC_ 55.4, ^2^*J*_HCC_ 4.8, ^3^*J*_HCCC_ 2.5); ^13^C-{^1^H} NMR spectrum (125.76 MHz, CD_3_CN) δ_C_ ppm (*J* Hz) 135.97 (d, C*^p^*, ^4^*J*_PCCCC_ 3.0), 134.99 (d, C^0^, ^2^*J*_PCC_ 10.3), 131.10 (d, C*^m^*, ^3^*J*_PCCC_ 12.8), 122.01 (q, CF_3_, ^1^*J*_FC_ 319.8), 120.05 (d, C*^i^*, ^1^*J*_PC_ 87.7), 66.58 (d, C^2^, ^2^*J*_PCC_ 5.2), 39.66 (d, C^3^, ^3^*J*_PCCC_ 17.6), 29.85 (d, C^1^, ^1^*J*_PC_ 55.6); ^31^P-{^1^H} NMR spectrum (CDCl_3_ + 10% acetone-*d*_6_): δ_P_ 24.5 ppm. MALDI-MS, *m*/*z* 399.5 [M − CF_3_SO_3_]^+^. Calcd. for C_21_H_21_O_2_P^+^: 400.2. Found: anal. C 48.15; H 3.88; Br 14.53%, calcd. for C_22_H_21_BrF_3_O_4_PS, C 48.10; H 3.85, Br 14.55%.

**(3-Fluoro-2-hydroxypropyl)triphenylphosphonium trifluoromethanesulfonate** (**3c**). Yield: 94%; mp 137 °C; IRS (cm^−1^): 3386, 3065, 2957, 2917, 1588, 1486, 1441, 1399, 1264, 1226, 1148, 1112, 1029, 996, 900, 831, 786, 745, 726, 691, 637, 574, 497; ^1^H NMR spectrum (400 MHz, CDCl_3_) δ ppm (*J* Hz) 7.81 (m, H*^p^*, 3H), 7.72 and 7.69 (two m, H*^o^*, H*^m^*, 12H), 4.50 (br d, H^3^, 2H, ^2^*J*_FH_ 48.5), 4.21 (m, H^2^, 1H), 3.89 (br s, OH, 1H), 3.72 (br ddd, H^1B^, 1H, ^2^*J*_H_^1A^_H_^1B^ 15.8, ^2^*J*_PH_^1B^ 12.9, ^3^*J*_H_^2^_H_^1B^ 8.7), 3.34 (br dd, H^1A^, 1H, ^2^*J*_H_^1B^_H_^1A^ 15.8, ^2^*J*_PH_^1A^ 12.1); ^13^C NMR spectrum (100.6 MHz, CDCl_3_) δ_C_ ppm (*J* Hz) 135.08 (br d (d), C*^p^*, ^1^*J*_HC_ 164.6, ^4^*J*_PCCCC_ 2.9), 133.86 (br dm (d), C*^o^*, ^1^*J*_HC_ 164.5, ^2^*J*_PCC_ 10.3), 130.37 (ddd (d), C*^m^*, ^1^*J*_HC_ 165.9, ^3^*J*_PCCC_ 12.7, ^3^*J*_HCCC_ 6.5), 120.51 (q (q), CF_3_, ^1^*J*_FC_ 320.1), 118.41 (dt (d), C*^i^*, ^1^*J*_PC_ 87.6, ^3^*J*_HCCC_ 7.3), 85.60 (dtm (dd), C^3^, ^1^*J*_FC_ 173.6, ^1^*J*_HC_ 160.3, ^3^*J*_PCCC_ 15.5), 65.84 (ddm (dd), C^2^, ^1^*J*_HC_ 147.4, ^2^*J*_FCC_ 20.5, ^2^*J*_PCC_ 5.6), 27.25 (tdm (dd), C^1^, ^1^*J*_HC_ 132.7, ^1^*J*_PC_ 56.7, ^3^*J*_FCCC_ 4.7); ^19^F NMR spectrum (376.5 MHz, CDCl_3_) δ_F_ −78.38 (s); ^31^P-{^1^H} NMR spectrum (242.94 MHz, CDCl_3_) δ_P_ 24.3 (s); MALDI-MS, *m*/*z* 339.5 [M − CF_3_SO_3_]^+^. Calcd. for C_21_H_21_O_2_P^+^: 339.4. Found: anal. C 53.87; H 4.20%, calcd. for C_22_H_21_F_4_O_4_PS, C 54.10; H 4.33%.

**(2-Hydroxy-3-methoxypropyl)triphenylphosphonium trifluoromethanesulfonate** (**3f**). Yield: 78%; mp 122–124 °C; IRS (KBr pellet, cm^−1^): 3399 br s (OH), 3064, 2996, 2960, 2923, 2910, 1588, 1486, 1459, 1439, 1416, 1382, 1337, 1282, 1249, 1225, 1196, 1161, 1110, 1046, 1026, 997, 969, 949, 869, 849, 843, 792, 750, 725, 715, 691, 638, 573, 540, 510, 497, 474; ^1^H NMR spectrum (400 MHz, CDCl_3_) δ ppm (*J* Hz) 7.72 (m, H*^p^*, 3H, ^3^*J*_HH_ 7.2, ^5^*J*_PH_ 2.0, ^4^*J*_HH_ 1.5), 7.67 (m, H*^o^*, 6H, ^3^*J*_PH_ 12.5, ^3^*J*_HH_ 7.2, ^4^*J*_HH_ 1.5), 7.67 (m, H*^m^*, 6H, ^3^*J*_HH_ 7.2, ^3^*J*_HH_ 7.2, ^4^*J*_PH_ 3.8), 5.85 (br s OH, 1H), 4.06 (m, H^2^, 1H), 3.26 (s, C^1′^, 3H), 3.54 (ddd, H^1B^, 1H, *B*-part of *ABX*-spectrum, ^2^*J*_H_^1A^_H_^1B^ 15.5, ^2^*J*_PH_^1B^ 13.0, ^2^*J*_H_^2^_H_^1B^ 10.4), 3.34 (ddd, H^1A^, 1H, *A*-part of *ABX*-spectrum, *J*_H_^1B^_H_^1A^ 15.5, ^2^*J*_PH_^1A^ 13.0, ^3^*J*_H_^2^_H_^1A^ 2.8), 3.49–3.50 (m, H^3^, 2H, *AB*-part of *ABX*-spectrum), 3.26 (s, H^1′^, 3H); ^13^C NMR spectrum (100.6 MHz, CDCl_3_) δ_C_ ppm (*J* Hz) 134.79 (dtd (d), C*^p^*, ^1^*J*_HC_ 163.8, ^3^*J*_HCCC_ 7.2, ^4^*J*_PCCCC_ 3.0), 133.56 (dddd (d), C*^o^*, ^1^*J*_HC_ 164.4, ^2^*J*_PCC_ 10.2, ^3^*J*_HCCC_ 7.4, ^3^*J*_HCCC_ 6.0–6.5), 130.10 (ddd (d), C*^m^*, ^1^*J*_HC_ 165.9, ^3^*J*_PCCC_ 12.8, ^3^*J*_HCCC_ 7.3), 120.37 (q (q), CF_3_, ^1^*J*_FC_ 320.2), 118.47 (dt (d), C*^i^*, ^1^*J*_PC_ 87.2, ^3^*J*_HCCC_ 8.0), 75.49 (tddt (d), C^3^, ^1^*J*_HC_ 142.0, ^3^*J*_PCCC_ 14.4, ^2^*J*_HCC_ 4.0, ^3^*J*_HCCC_ 2.0–2.2), 65.01 (dm (d), C^2^, ^1^*J*_HC_ 144.3, ^2^*J*_PCC_ 5.7, ^2^*J*_HCC_ 3.0–4.0, ^2^*J*_HCC_ 3.0–4.0), 58.88 (qt (s), C^1′^, ^1^*J*_HC_ 141.7, ^3^*J*_HCOC_ 2.6), 27.83 (tdm (d), C^1^, ^1^*J*_HC_ 130.4, ^1^*J*_PC_ 55.0, ^2^*J*_HCC_ 4.5, ^3^*J*_HCCC_ 1.7–2.0); ^19^F NMR spectrum (376.5 MHz, CDCl_3_) δ_F_ −78.33 (s); ^31^P-{^1^H} NMR spectrum (162.0 MHz, CDCl_3_) δ_P_ 24.7 (s); MALDI-MS, *m*/*z* 351.1 [M − CF_3_SO_3_]^+^. Calcd. for C_22_H_24_O_2_P^+^: 351.4. Found: anal. C 55.16; H 4.73%, calcd. for C_23_H_24_F_3_O_5_PS, C 55.20; H 4.83%.

**(3-Ethoxy-2-hydroxypropyl)triphenylphosphonium trifluoromethanesulfonate** (**3g**). Yield: 77%; mp 99–101 °C; IRS (KBr pellet, cm^−1^): 3386 br s (OH), 3058, 2973, 2922, 2892, 2802, 1589, 1486, 1440, 1411, 1375, 1357, 1286, 1247, 1226, 1158, 1110, 1061, 1029, 997, 949, 916, 869, 838, 793, 751, 724, 716, 690, 639, 573, 540, 509, 496, 476, 421; ^1^H NMR spectrum (400 MHz, CDCl_3_) δ ppm (*J* Hz) 7.76 (m, H*^p^*, 3H, ^3^*J*_HH_ 7.2, ^5^*J*_PH_ 2.1, ^4^*J*_HH_ 1.5), 7.70 (m, H*^o^*, 6H, ^3^*J*_PH_ 12.5, ^3^*J*_HH_ 7.2, ^4^*J*_HH_ 1.5), 7.65 (m, H*^m^*, 6H, ^3^*J*_HH_ 7.2, ^3^*J*_HH_ 7.2, ^4^*J*_PH_ 3.6), 5.08 (br s, OH, 1H), 4.10 (m, H^2^, 1H, ^3^*J*_PH_^2^ 8.7, ^3^*J*_HH_^2^ 5.9, ^3^*J*_H_^1A^_H_^2^ 2.8), 3.40 (ddd, H^1A^, 1H, *A*-part of *ABX*-spectrum, ^2^*J*_H_^1B^_H_^1A^ 15.8, ^2^*J*_PH_^1A^ 13.2, ^3^*J*_H_^2^_H_^1A^ 2.8), 3.56–3.58 (m, H^3^, H^1B^, 3H), 3.47 and 3.50 (two m, H^1′^, 2H, *AB*-part of *ABX*_3_-spectrum, ^2^*J*_H_^1′B^_H_^1′A^ 11.0, ^3^*J*_H_^2′^_H_^1′^ 7.0), 1.13 (t, H^2′^, 3H, ^3^*J*_H_^1′^_H_^2′^ 7.0); ^13^C NMR spectrum (100.6 MHz, CDCl_3_) δ_C_ ppm (*J* Hz) 134.86 (dtd (d), C*^p^*, ^1^*J*_HC_ 163.5, ^3^*J*_HCCC_ 7.2, ^4^*J*_PCCCC_ 3.0), 133.64 (dddd (d), C*^o^*, ^1^*J*_HC_ 164.3, ^2^*J*_PCC_ 10.2, ^3^*J*_HCCC_ 7.3, ^3^*J*_HCCC_ 6.5), 130.16 (ddd (d), C*^m^*, ^1^*J*_HC_ 166.0, ^3^*J*_PCCC_ 12.5, ^3^*J*_HCCC_ 7.2), 120.47 (q (q), CF_3_, ^1^*J*_FC_ 320.2), 118.55 (dt (d), C*^i^*, ^1^*J*_PC_ 87.2, ^3^*J*_HCCC_ 8.0), 73.52 (tdm (d), C^3^, ^1^*J*_HC_ 143.5, ^3^*J*_PCCC_ 14.7), 66.73 (tm (s), C^1′^, ^1^*J*_HC_ 140.4), 65.19 (ddm (d), C^2^, ^1^*J*_HC_ 146.3, ^2^*J*_PCC_ 5.7, ^2^*J*_HCC_ 4.0–4.2, ^2^*J*_HCC_ 4.0–4.2), 28.03 (tdm (d), C^1^, ^1^*J*_HC_ 132.0, ^1^*J*_PC_ 54.9, ^2^*J*_HCC_ 4.0–4.2, ^3^*J*_HCCC_ 2.8–3.0), 14.93 (qt (s), C^2′^, ^1^*J*_HC_ 126.1, ^2^*J*_HCC_ 2.8-3.0); ^19^F NMR spectrum (376.5 MHz, CDCl_3_) δ_F_ −78.31 (s); ^31^P-{^1^H} NMR spectrum (162.0 MHz, CDCl_3_) δ_P_ 24.1 (s); MALDI-MS, *m*/*z* 365.6 [M − CF_3_SO_3_]^+^. Calcd. for C_23_H_26_O_2_P^+^: 365.4. Found: anal. C 55.89; H 4.84%, calcd. for C_24_H_26_F_3_O_5_PS, C 56.03; H 5.09%.

**(2-Hydroxy-3-(2,2,3,3-tetrafluoropropoxy)propyl)triphenylphosphonium trifluoromethanesulfonate** (**3h**). Yield: 65%; Oil; IRS (film, cm^−1^): 3397 br s (OH), 3066, 2919, 1589, 1487, 1440, 1400, 1339, 1287, 1257, 1226, 1203, 1163, 1111, 1030, 998, 939, 886, 832, 787, 748, 722, 691, 639, 574, 543, 515, 443; ^1^H NMR spectrum (400 MHz, CDCl_3_) δ ppm (*J* Hz) 7.77 (m, H*^p^*, 3H, ^3^*J*_HH_ 7.2, ^5^*J*_PH_ 1.8, ^4^*J*_HH_ 1.5–1.6), 7.68 (m, H*^o^*, 6H, ^3^*J*_PH_ 12.6, ^3^*J*_HH_ 7.2, ^4^*J*_HH_ 1.6), 7.65 (m, H*^m^*, 6H, ^3^*J*_HH_ 7.2, ^3^*J*_HH_ 7.2, ^4^*J*_PH_ 3.7), 5.94 (tt, H^3′^, 1H, ^2^*J*_FH_ 53.0, ^3^*J*_FH_ 5.2), 4.40 (br s, OH, 1H), 4.13 (br m, H^2^, 1H), 3.88 and 3.85 (two m, H^1′^, 2H, *AB*-part of *ABX*_2_-spectrum, ^3^*J*_FH_^A,B^ 12.4, ^2^*J*_H_^1′A^_H_^1′B^ 12.2), 3.75 (m, H^3^, 2H, *AB*-part of *ABX*-spectrum, ^2^*J*_H_^3A^_H_^3B^ 10.7), 3.57 (ddd, H^1B^, 1H, *B*-part of *ABX*-spectrum, ^2^*J*_H_^1B^_H_^1A^ 15.7, ^2^*J*_PH_^1B^ 11.2, ^3^*J*_H_^2^_H_^1A^ 10.5), 3.35 (ddd, H^1A^, 1H, *A*-part of *ABX*-spectrum, ^2^*J*_H_^1B^_H_^1A^ 15.7, ^2^*J*_PH_^1A^ 11.2, ^3^*J*_H_^2^_H_^1A^ 2.5); ^13^C NMR spectrum (100.6 MHz, CDCl_3_) δ_C_ ppm (*J* Hz) 134.92 (dtd (d), C*^p^*, ^1^*J*_HC_ 163.7, ^3^*J*_HCCC_ 7.0, ^4^*J*_PCCCC_ 3.0), 133.68 (dddd (d), C*^o^*, ^1^*J*_HC_ 164.6, ^2^*J*_PCC_ 10.2, ^3^*J*_HCCC_ 7.6, ^3^*J*_HCCC_ 6.3), 130.22 (ddd (d), C*^m^*, ^1^*J*_HC_ 166.1, ^3^*J*_PCCC_ 12.8, ^3^*J*_HCCC_ 7.2), 120.50 (q (q), CF_3_, ^1^*J*_FC_ 320.2), 118.50 (dt (d), C*^i^*, ^1^*J*_PC_ 87.5, ^3^*J*_HCCC_ 8.0), 114.96 (ttdt (tt), C^2′^, ^1^*J*_FC_ 250.0, ^2^*J*_FCC_ 26.6, ^2^*J*_HC_^3′^_C_ 5.7, ^2^*J*_HC_^3′^_C_ 1.6), 109.25 (tdtm (tt), C^3′^, ^1^*J*_FC_ 249.1, ^1^*J*_HC_ 193.2, ^2^*J*_FCC_ 34.2, ^3^*J*_HC_^1′^_CC_ 1.8), 75.50 (tdm (d), C^3^, ^1^*J*_HC_ 142.7, ^3^*J*_PCCC_ 5.2), 68.34 (ttt (t), C^1′^, ^1^*J*_HC_ 147.0, ^2^*J*_FCC_ 8.1, ^2^*J*_FCCC_ 3.2), 65.15 (dm (d), C^2^, ^1^*J*_HC_ 146.5, ^2^*J*_PCC_ 5.6, ^2^*J*_HCC_ 2.2–2.5), 27.60 (tdm (d), C^1^, ^1^*J*_HC_ 131.8, ^1^*J*_PC_ 55.8); ^19^F NMR spectrum (376.5 MHz, CDCl_3_, δ_F_ ppm, *J* Hz): −78.44 ((s), CF_3_ 3F), −127.97 (tdt, C^2′^F_2_, 2F, ^3^*J*_H_^1′^_F_^2′^ 12.8, ^3^*J*_H_^3′^_F_^2′^ 5.4, ^3^*J*_F_^3′^_F_^2′^ 5.5), −132.20 (m, F^3′B^, 1F, ^2^*J*_F_^3′A^_F_^3′B^ 450.0, ^2^*J*_H_^3′^_F_^3′B^ 52.9), −139.60 (m, F^3′A^, 1F, ^2^*J*_F_^3′B^_F_^3′A^ 450.0, ^2^*J*_H_^3′^_F_^3′B’^ 53.0, ^3^*J*_F_^2′^_F_^3′A^ 5.5, ^4^*J*_H_^1′^_F_^3′A^ 1.5); ^31^P-{^1^H} NMR spectrum (162.0 MHz, CDCl_3_) δ_P_ 24.1 (s); ^31^P-{^1^H} NMR spectrum (162.0 MHz, CH_3_CN) δ_P_ 25.1 (s); MALDI-MS, *m*/*z* 451.7 [M − CF_3_SO_3_]^+^. Calcd. for C_24_H_24_F_4_O_2_P^+^: 451.4. Found: anal. C 49.75; H 3.84%, calcd. for C_25_H_24_F_7_O_5_PS, C 50.01; H 4.03%.

**(3-(Allyloxy)-2-hydroxypropyl)triphenylphosphonium trifluoromethanesulfonate** (**3i**). Yield: 87%; mp 124–125 °C (from the CH_2_Cl_2_-benzene mixture); IRS (cm^−1^): 3370–3390 br s (ν_OH_), 1615, 1590, 1488, 1440, 1355, 1340, 1305, 1290, 1250, 1230, 1175, 1160, 1115, 1054, 1033, 1001, 940, 890, 855, 845, 796, 755, 726, 717, 695, 643, 575, 541, 515, 500, 480; ^1^H NMR spectrum (250 MHz, CD_3_OD) δ ppm (*J* Hz) 7.90–8.02 (m, C_6_H_5_); 6.11 (ddt, =CH, ^3^*J*_H_^X^_H_ 17.3, ^3^*J*_H_^A^_H_ 10.4, ^3^*J*_HH_ 5.6), 5.47 (ddt, =CH, ^3^*J*_HH_^X^ 17.3, ^2^*J*_H_*^gem^*_H_ 1.6, ^4^*J*_HH_ 1.5), 5.36 (ddt, =CH, ^3^*J*_H_^A^_H_ 10.4, ^2^*J*_H_*^gem^*_H_ 1.6, ^4^*J*_HH_ 1.3), 4.18 (m, CH–O, ^3^*J*_HH_ 5.5), 3.95 (ddd, PCH^B^, ^2^*J*_H_^A^_H_^B^ 15.5, ^2^*J*_PH_^B^ 11.5, ^3^*J*_HH_^B^10.6), 3.74 (m, =CCH_2_, ^3^*J*_HH_ 5.6, ^4^*J*_H_^A^_H_ 1.5, ^4^*J*_H_^X^_H_ 1.3), 3.69 (ddd, PCH^A^, ^2^*J*_H_^B^_H_^A^ 15.5, ^2^*J*_PH_^A^ 13.8, ^3^*J*_HH_^A^ 2.7); ^1^H NMR spectrum (600 MHz, CDCl_3_) δ ppm (*J* Hz) 7.74 (m, H*^p^*, 3H, ^3^*J*_HH_ 7.4, ^5^*J*_HH_ 1.2), 7.67 (m, H*^o^*, 6H, ^3^*J*_PH_ 12.8, ^3^*J*_HH_ 7.4), 7.63 (m, H*^m^*, 6H, ^3^*J*_HH_ 7.4, ^4^*J*_PH_ 3.6), 5.81 (ddt, H^2′^, 1H, ^3^*J*_HH_ 17.2, ^3^*J*_HH_ 10.4, ^3^*J*_HH_ 5.6), 5.18 (dd, H^3′X^, 1H, ^3^*J*_HH_ 17.2, ^2^*J*_HH_ 1.2), 5.10 (dd, H^3′A^, 1H, ^3^*J*_HH_ 10.4, ^2^*J*_HH_ 1.2), 4.10 (br m, H^2^, OH, 2H), 3.93 (m, H^1′^, 2H, *AB*-part of *ABX*-spectrum, ^2^*J*_H1′AH1′B_ 12.7, ^3^*J*_H2H1′_ 5.6), 3.56 (m, H^1B^, 1H, ^2^*J*_PH1B_ 12.0, ^2^*J*_H1AH1B_ 15.7, ^3^*J*_H2H1B_ 10.4), 3.37 (ddd, H^1A^, 1H, ^3^*J*_H1BH1A_ 15.7, ^2^*J*_PH1A_ 12.0, ^3^*J*_H2H1A_ 2.4); ^13^C NMR spectrum (150.9 MHz, CDCl_3_) δ_C_ ppm (*J* Hz) 134.91 (dtd (d), C*^p^*, ^1^*J*_HC_ 164.2, ^3^*J*_HCCC_ 7.3, ^4^*J*_PCCCC_ 3.1), 134.23 (ddt (s), C^2′^, ^1^*J*_HC_ 155.5, ^2^*J*_HCC_ 3.8, ^2^*J*_HCC_ 3.8), 133.71 (dddd (d), C*^o^*, ^1^*J*_HC_ 164.6, ^2^*J*_PCC_ 10.2, ^3^*J*_HCCC_ 7.8, ^3^*J*_HCCC_ 6.7), 130.23 (ddd (d), C*^m^*, ^1^*J*_HC_ 166.4, ^3^*J*_PCC_ 12.5, ^3^*J*_HCCC_ 7.4), 120.64 (q (q), CF_3_, ^1^*J*_FC_ 320.6), 118.67 (dt (d), C*^i^*, ^1^*J*_PC_ 187.1, ^3^*J*_HCCC_ 8.4), 117.39 (ddt (s), C^3′^, ^1^*J*_HC_ 158.8, ^1^*J*_HC_ 155.2, ^3^*J*_HCCC_ 5.6), 73.37 (tdm (d), C^3^, ^1^*J*_HC_ 144.5, ^3^*J*_PCCC_ 14.5), 72.28 (tm (s), C^1′^, ^1^*J*_HC_ 141.8, ^3^*J*_HCCC_ 12.9, ^3^*J*_HCCC_ 6.9, ^2^*J*_HCC_ 4.4, ^3^*J*_HCOC_ 2.9), 65.28 (dm (d), C^2^, ^1^*J*_HC_ 147.1, ^2^*J*_PCC_ 5.8, ^2^*J*_HCC_ 2.8), 28.24 (br td (d), C^1^, ^1^*J*_HC_ 132.2, ^1^*J*_PC_ 54.1); ^19^F NMR spectrum (376.5 MHz, CDCl_3_) δ_F_ −78.29 (s); ^31^P-{^1^H} NMR spectrum (162.0 MHz, CH_2_Cl_2_) δ_P_ 23.5; ^31^P-{^1^H} NMR spectrum (242.94 MHz, CDCl_3_) δ_P_ 24.2; MALDI-MS, *m*/*z* 377.4 [M − CF_3_SO_3_]^+^. Calcd. for C_24_H_26_O_2_P^+^: 377.4. Found: anal. C 56.95; H 4.93%, calcd. for C_25_H_26_F_3_O_5_PS, C 57.03; H 4.98%.

**(3-(Decyloxy)-2-hydroxypropyl)triphenylphosphonium trifluoromethanesulfonate** (**3j**). Yield: 75%; mp 78–79 °C; IRS (cm^−1^): 3381 br s (OH), 3066, 2927, 2856, 2795, 1486, 1467, 1459, 1440, 1406, 1363, 1287, 1246, 1227, 1162, 1111, 1031, 997, 950, 879, 838, 791, 748, 724, 715, 689, 640, 573, 530, 510, 497, 465; ^1^H NMR spectrum (400 MHz, CDCl_3_) δ ppm (*J* Hz) 7.75 (m, H*^p^*, 3H, ^3^*J*_HH_ 7.2, ^5^*J*_PH_ 2.0, ^4^*J*_HH_ 1.5), 7.68 (m, H*^o^*, 6H, ^3^*J*_PH_ 12.5, ^3^*J*_HH_ 7.2, ^4^*J*_HH_ 1.5), 7.64 (m, H*^m^*, 6H, ^3^*J*_HH_ 7.2, ^4^*J*_PH_ 3.6), 4.08 (m, H^2^, 1H), 3.98 (br s, OH, 1H), 3.54–3.55 (m, H^1B^, H^3^, 3H), 3.37–3.40 (m, H^1A^, H^1′^, 3H), 1.49 (m, H^2′^, 2H, ^3^*J*_HH_ 6.8–6.9), 1.22 (m, H^3′^-H^9′^, 14H), 0.84 (t, H^10′^, 3H, ^3^*J*_HH_ 7.0); ^13^C NMR spectrum (100.6 MHz, CDCl_3_) δ_C_ ppm (*J* Hz) 134.91 (dtd (d), C*^p^*, ^1^*J*_HC_ 163.7, ^3^*J*_HCCC_ 7.1, ^4^*J*_PCCCC_ 3.0), 133.71 (dddd (d), C*^o^*, ^1^*J*_HC_ 165.1, ^2^*J*_PCC_ 10.3, ^3^*J*_HCCC_ 7.4, ^3^*J*_HCCC_ 6.4), 130.22 (ddd (d), C*^m^*, ^1^*J*_HC_ 166.0, ^3^*J*_PCCC_ 12.7, ^3^*J*_HCCC_ 7.2), 120.58 (q (q), CF_3_, ^1^*J*_FC_ 320.4), 118.63 (dt (d), C*^i^*, ^1^*J*_PC_ 87.1, ^3^*J*_HCCC_ 8.0), 73.79 (tdm (d), C^3^, ^1^*J*_HC_ 142.3, ^3^*J*_PCCC_ 14.6), 71.61 (tm (s), C^1′^, ^1^*J*_HC_ 141.6), 65.26 (dm (d), C^2^, ^1^*J*_HC_ 146.2, ^2^*J*_PCC_ 5.7, ^2^*J*_HCC_ 2.5–3.3), 32.84 (tm (s), C^2′^, ^1^*J*_HC_ 125.7), 29.58 (tm (s), C^3′^, ^1^*J*_HC_ 123.0–124.0), 29.51 (tm (s), C^4′^, C^5′^, ^1^*J*_HC_ 124.0–125.0), 29.41 (tm (s), C^6′^, ^1^*J*_HC_ 124.0–125.0), 29.27 (tm (s), C^7′^, ^1^*J*_HC_ 123.0–124.0), 28.15 (tdm (d), C^1^, ^1^*J*_HC_ 134.0, ^1^*J*_PC_ 54.6), 26.05 (tm (s), C^8′^, ^1^*J*_HC_ 124.0–125.0), 22.63 (tm (s), C^9′^, ^1^*J*_HC_ 124.3), 14.10 (qm (s), C^10′^, ^1^*J*_HC_ 125.0, ^3^*J*_HCCC_ 2.5–2.7, ^2^*J*_HCC_ 2.5); ^19^F NMR spectrum (376.5 MHz, CDCl_3_) δ_F_ −78.28 (s); ^31^P-{^1^H} NMR spectrum (162.0 MHz, CDCl_3_) δ_P_ 24.2 (s); MALDI-MS, *m*/*z* 477.4 [M − CF_3_SO_3_]^+^. Calcd. for C_31_H_42_O_2_P^+^: 477.7. Found: anal. C 61.16; H 6.54%, calcd. for C_32_H_42_F_3_O_5_PS, C 61.33; H 6.76%.

**(2-Hydroxy-3-palmitoyloxy-propyl)triphenylphosphonium t****rifluoromethanesulfonate** (**3k**). To a solution of glycidyl palmitate (for the synthesis and spectral data see [203]) (0.512 g, 1.64 mmol) in dry CHCl_3_ (50 mL) was added a solution of triphenylphosphonium triflate in CHCl_3_ (1.9 mL, 1.27 mmol, C = 0.2748 g/mL) in an argon atmosphere. The reaction mixture was stirred at room temperature for 3 h. The solvent was completely removed from the reaction medium in vacuo. The product was purified by filtration through a layer of silica gel, which was initially washed with a mixture of petroleum ether/ethyl acetate 3:1 (*v*/*v*), and then with ethanol. Yield: (95%); colorless amorphous powder; mp 40–41 °C; IRS (KBr, cm^−1^): 3380, 3063, 2018, 1938, 1909, 1830, 1737, 1197, 1166, 1031; ^1^H NMR spectrum (400 MHz, CDCl_3_) δ 7.77, 7.69, 7.65 (three m, 15H, H*^p^*, H*^o^*, H*^m^*, ^3^*J*_HH_ 7.6, ^3^*J*_PH_ 12.8, ^4^*J*_PH_ 3.3), 4.68 (brs, 1H, H^2^), 4.20 (br. s, 2H, H^3^), 3.64 (br. m, 1H, H^1B^, ^2^*J*_HH_ 15.0, ^2^*J*_PH_ 14.0), 3.32 (br. d. d, 1H, H^1A^, ^2^*J*_HH_ 15.0, ^2^*J*_PH_ 14.0), 2.26 (m, 2H, CH_2_C(O), ^3^*J*_HH_7.2), 1.54 (m, 2H, ^3^*J*_HH_ 7.2–7.4), 1.24 (br. s, 24H), 0.86 (t, 3H, ^3^*J*_HH_ 7.0); ^13^C–{^1^H} NMR spectrum (100.6 MHz, CDCl_3_) δ_C_ 173.83 (s, C(O)O), 135.04 (d, C^p^, ^4^*J*_PC_ 3.0), 133.85 (s, C^o^, ^3^*J*_PC_ 10.3), 130.34 (d, C^m^, ^3^*J*_PC_ 12.8), 118.57 (q, CF_3_, ^1^*J*_FC_ 320.0), 118.57 (d, C^i^, ^1^*J*_PC_ 87.4), 67.61 (d, C^3^, ^3^*J*_PC_ 15.9), 64.65 (d, C^2^, ^2^*J*_PC_ 5.4), 34.02 (s, C^1′^), 31.97 (s, C^2′^), 29.75 (brs, C^3′^, C^4′^, C^5′^), 29.72 (s, C^6′^), 29.70 (s, C^7′^), 29.69 (s, C^8′^), 2954 (s, C^9′^), 29.41 (s, C^10′^) 29.38 (s, C^11′^), 29.16 (s, C^12′^), 28.32 (d, C^1^, ^1^*J*_PC_ 55.6), 24.85 (s, C^13′^), 22.74 (s, C^14′^), 14.19 (s, C^15′^); ^31^P–{^1^H} NMR spectrum (242.9 MHz, CDCl_3_) δ_P_ 24.2 (s); MALDI-MS, *m*/*z* 575.3 [M − CF_3_SO_3_]^+^. Calcd. for C_37_H_52_O_3_P^+^: 575.4. Found: anal. C 62.59; H 7.28%, calcd. for C_38_H_52_F_3_O_6_PS, C 62.97; H 7.23%.

**(2-Hydroxy-3-stearoyloxy-propyl)triphenylphosphonium****trifluoromethanesulfonate** (**3l**). To a solution of glycidyl stearate (for the synthesis and spectral data see [203]) (0.5 g, 1.47 mmol) in dry CHCl_3_ (50 mL) was added a solution of triphenylphosphonium triflate in CHCl_3_ (1.85 mL, 1.23 mmol, C = 0.2748 g/mL) in an argon atmosphere. The reaction mixture was stirred at room temperature for 3 h. The solvent was completely removed from the reaction medium in vacuo. The product was purified by filtration through a layer of silica gel, which was initially washed with a mixture of petroleum ether/ethyl acetate 3:1 (*v*/*v*), and then with ethanol. Yield: 37%; colorless amorphous powder; mp = 60–61 °C; IRS (KBr, cm^−1^) 3436, 3063, 2018, 1938, 1909, 1830, 1737, 1160, 1113, 1031; ^1^H NMRspectrum (400 MHz, CDCl_3_) δ 7.77, 7.71, 7.68 (three m, 15H, H*^p^*, H*^o^*, H*^m^*), 4.52 (brs, 1H, OH), 4.22 (br. s., 1H, H^2^), 4.10–4.16 (m, 2H, H^3^), 3.63 (br. m, 1H, H^1B^, ^2^*J*_HH_ 12.9, ^2^*J*_PH_ 11.1), 3.32 (br. d. d, 1H, H^1A^, ^2^*J*_HH_ 12.9, ^2^*J*_PH_ 11.1), 2.30 (t, 2H, CH_2_C(O), ^3^*J*_HH_ 7.5), 1.55 (m, 2H, ^3^*J*_HH_ 7.2–7.4), 1.24 (br. s, 28H), 0.86 (t, 3H, ^3^*J*_HH_ 7.0); ^13^C–{^1^H} NMRspectrum (100.6 MHz, CDCl_3_) δ_C_ 173.95 (s, C(O)O), 134.82 (d, C^p^, ^4^*J*_PC_ 2.9), 133.78 (s, C^o^, ^3^*J*_PC_ 10.2), 130.20 (d, C^m^, ^3^*J*_PC_ 12.7), 120.43 (q, CF_3_, ^1^*J*_FC_ 320.0), 118.51 (d, C^i^, ^1^*J*_PC_ 87.4), 67.65 (d, C^3^, ^3^*J*_PC_ 16.6), 64.69 (d, C^2^, ^2^*J*_PC_ 5.7), 34.11 (s, C^1′^), 31.93 (s, C^2′^), 29.71 (brs, C^3′^, C^4′^, C^5′^, C^6′^), 29.67 (s, C^7′^, C^8′^, C^9′^), 29.62 (s, C^10′^), 29.47 (s, C^11′^), 29.37 (s, C^12′^), 29.67 (s, C^13′^), 29.13 (s, C^14′^), 29.42 (d, C^1^, ^1^*J*_PC_ 54.5), 24.89 (s, C^15′^), 22.70 (s, C^16′^), 14.13 (s, C^17′^);^31^P–{^1^H} NMRspectrum (242.9 MHz, CDCl_3_) δ_P_ 24.3 (s); MALDI-MS, *m*/*z* 603.4 [M − CF_3_SO_3_]^+^. Calcd. for C_39_H_56_O_3_P^+^: 603.9. Found: anal. C 63.74; H 7.57%, calcd. for C_40_H_56_F_3_O_6_PS, C 63.81; H 7.50%.

#### 3.2.5. General Procedure for the Synthesis of (3-Alkoxy-2-hydroxypropyl)triphenylphosphonium Iodide (**4a**–**d**)

Triphenylphosphine (4–6 mmol) was added to the solution of iodide **6a–d** (1 mmol) in dry acetonitrile under argon, and the mixture was stirred under reflux for 10 h. Acetonitrile was removed under reduced pressure, and the precipitate was washed with ether (3 × 5 mL). The resulting precipitate was dried in *vacuo* to give the pure TPP conjugate °C; IRS (KBr pellet, cm^−1^): 3245 very br. s. (OH), 3051, 2982, 2931, 2876, 2828, 1585, 1482, 1461, 1437, 1413, 1390, 1301, 1239, 1196, 1153, 1121, 1110, 1044, 1029, 997, 868, 833, 791, 751, 719, 694, 588, 532, 500, 470, 448. ^1^H NMR spectrum (400 MHz, CDCl_3_) δ ppm (*J* Hz) 7.75 (m, H*^p^*, 3H, ^3^*J*_HH_ 7.2, ^5^*J*_PH_ 2.0, ^4^*J*_HH_ 1.5), 7.72 (m, H*^o^*, 6H, ^3^*J*_PH_ 12.2, ^3^*J*_HH_ 7.2, ^4^*J*_HH_ 1.5), 7.63 (m, H*^m^*, 6H, ^3^*J*_HH_ 7.2, ^4^*J*_PH_ 3.5), 3.72 (very br s, OH, 1H), 4.22 (m, H^2^, 1H, 10.0, ^3^*J*_H_^3A^_H_^2^ 7.1, ^3^*J*_H_^3B^_H_^2^ 6.1, ^3^*J*_H_^1A^_H_^2^ 2.8), 3.74 (ddd, H^1B^, 1H, *B*-part of *ABX*-spectrum, ^2^*J*_H_^1A^_H_^1B^ 15.6, ^2^*J*_PH_^1B^ 12.0, ^3^*J*_H_^2^_H_^1B^ 10.0), 3.59 (ddd, H^1A^, 1H, *A*-part of *ABX*-spectrum, ^2^*J*_H_^1B^_H_^1A^ 15.6, ^2^*J*_PH_^1A^ 12.0, ^3^*J*_H_^2^_H_^1A^ 2.8), 3.62–3.68 (m, H^3^, 2H, *AB*-part of *ABXY*-spectrum, ^2^*J*_H_^3A^_H_^3B^ 9.6, ^3^*J*_H_^2^_H_^3A^ 7.1, ^3^*J*_H_^2^_H_^3B^ 6.1, ^4^*J*_PH_^1A^ 2.2, ^4^*J*_PH_^1B^ 1.1), 3.26 (s, H^1′^, 3H), ^13^C NMR spectrum (100.6 MHz, CDCl_3_) δ_C_ ppm (*J* Hz) 134.63 (dtd (d), C*^p^*, ^1^*J*_HC_ 163.8, ^3^*J*_HCCC_ 7.1, ^4^*J*_PCCCC_ 3.0), 133.55 (dddd (d), C*^o^*, ^1^*J*_HC_ 164.5, ^2^*J*_PCC_ 10.3, ^3^*J*_HCCC_ 7.3, ^3^*J*_HCCC_ 6.7), 129.99 (ddd (d), C*^m^*, ^1^*J*_HC_ 165.8, ^3^*J*_PCCC_ 12.7, ^3^*J*_HCCC_ 7.3), 118.19 (dt (d), C*^i^*, ^1^*J*_PC_ 87.2, ^3^*J*_HCCC_ 8.4), 75.31 (tdm (d), C^3^, ^1^*J*_HC_ 143.0, ^3^*J*_PCCC_ 14.0, ^2^*J*_HCC_ 4.8–5.0, ^3^*J*_HCCC_ 1.8–2.0), 64.56 (dt (d), C^2^, ^1^*J*_HC_ 146.6, ^2^*J*_PCC_ 5.6, ^2^*J*_HCC_ 2.8–3.1, ^2^*J*_HCC_ 2.8–3.1), 58.78 (qt (s), C^1′^, ^1^*J*_HC_ 141.6, ^3^*J*_HCOC_ 2.7), 27.87 (tdm, C^1^, ^1^*J*_HC_ 131.3, ^1^*J*_PC_ 54.4, ^2^*J*_HCC_ 3.5–4.0, ^3^*J*_HCCC_ 2.6); ^31^P-{^1^H} NMR spectrum (162.0 MHz, CDCl_3_) δ_P_ 24.7 (s); MALDI-MS, *m*/*z* 351.4 [M − I]^+^. Calcd. for C_22_H_24_O_2_P^+^351.2. Found: anal. C 55.19; H 4.96; I 26.50%, calcd. for C_22_H_24_IO_2_P, C 55.24; H 5.06; I 26.53%.

**(3-Ethoxy-2-hydroxypropyl)triphenylphosphonium iodide** (**4b**). Yield: 95%; Oil; IRS (film, cm^−1^): 3306 very br s (OH), 3055, 2974, 2870, 1615, 1588, 1485, 1438, 1384, 1337, 1316, 1271, 1238, 1181, 1114, 1086, 1027, 1014, 997, 935, 907, 868, 825, 783, 748, 722, 692, 618, 601, 542, 507, 444. ^1^H NMR spectrum (400 MHz, CDCl_3_) δ ppm (*J* Hz): 7.82 (m, H*^p^*, 3H, ^3^*J*_HH_7.2, ^5^*J*_PH_ 2.0, ^4^*J*_HH_ 1.5), 7.79 (m, H*^o^*, 6H, ^3^*J*_PH_ 15.3, ^3^*J*_HH_ 7.2, ^4^*J*_HH_ 1.5), 7.68 (m, H*^m^*, 6H, ^3^*J*_HH_ 7.2, ^3^*J*_HH_ 7.2, ^4^*J*_PH_ 3.3), 4.26 (m, H^2^, 1H, ^3^*J*_H_^1B^_H_^2^ 10.3, ^3^*J*_H_^1A^_H_^2^ 2.8, ^3^*J*_H_^3A^_H_^2^ 2.5), 3.69 (ddd, H^1A^, 1H, *A*-part of *ABX*-spectrum, ^2^*J*_H_^1B^_H_^1A^ 15.3, ^2^*J*_PH_^1A^ 12.0, ^3^*J*_H_^2^_H_^1A^ 2.8), 3.74 and 3.80 (two m, H^3^, 2H, *AB*-part of *ABX*-spectrum, ^2^*J*_H_^3B^_H_^3A^ 9.5, ^3^*J*_H_^2^_H_^3B^ 6.6, ^3^*J*_H_^2^_H_^3A^ 2.5), 3.88 (ddd, H^1B^, 1H, *B*-part of *ABX*-spectrum, ^2^*J*_H_^1B^_H_^1A^ 15.3, ^2^*J*_PH_^1B^ 12.0, ^3^*J*_H_^2^_H_^1B^ 10.3), 3.52–3.56 (m, H^1′^, 2H, *AB*-part of *ABX*_3_-spectrum, ^2^*J*_H_^1′B^_H_^1′A^ 10.5, ^3^*J*_H_^2′^_H_^1A’^ 7.1, ^3^*J*_H_^2′^_H_^1B’^ 7.1), 1.16 (t, H^2′^, 3H, ^3^*J*_H_^1′^_H_^2′^ 7.0–7.1); ^13^C NMR spectrum (100.6 MHz, CDCl_3_) δ_C_ ppm (*J* Hz): 134.87 (dtd (d), C*^p^*, ^1^*J*_HC_ 163.7, ^3^*J*_HCCC_ 7.2, ^4^*J*_PCCCC_ 3.0), 133.89 (dddd (d) C*^o^*, ^1^*J*_HC_ 164.7, ^2^*J*_PCC_ 10.2, ^3^*J*_HCCC_ 7.6, ^3^*J*_HCCC_ 6.8), 130.23 (ddd (d), C*^m^*, ^1^*J*_HC_ 165.7, ^3^*J*_PCCC_ 12.8, ^3^*J*_HCCC_ 7.2), 118.65 (dt (d), C*^i^*, ^1^*J*_PC_ 87.1, ^3^*J*_HCCC_ 8.4), 73.68 (tdm (d), C^3^, ^1^*J*_HC_ 145.0, ^3^*J*_PCCC_ 14.5, ^2^*J*_HCC_ 3.4–3.5, ^3^*J*_HCCC_ 2.0–2.6), 66.76 (tm (s), C^1′^, ^1^*J*_HC_ 141.6, ^2^*J*_HCC_ 4.6, ^3^*J*_HCCC_ 2.1–2.2), 65.08 (ddm, C^2^, ^1^*J*_HC_ 147.5, ^2^*J*_PCC_ 5.7), 28.36 (tdm, C^1^, ^1^*J*_HC_ 130.0, ^1^*J*_PC_ 54.2), 15.14 (qt (s), C^2′^, ^1^*J*_HC_ 126.0, ^2^*J*_HCC_ 2.5); ^31^P-{^1^H} NMR spectrum (162.0 MHz, CDCl_3_): δ_P_ 25.3 (s); MALDI-MS, *m*/*z* 365.5 [M − I]^+^. Calcd. for C_23_H_26_O_2_P^+^: 365.4. Found: anal. C 56.05; H 5.27; I 25.70%, calcd. for C_23_H_26_IO_2_P, C 56.11; H 5.32; I 25.78%.

**(3-(Allyloxy)-2-hydroxypropyl)triphenylphosphonium iodide** (**4c**). Yield is 90%; IRS (film, cm^−1^): 3305, 3055, 2865, 1485, 1438, 1339, 1111, 997, 931, 826, 748, 721, 690; ^1^H NMR spectrum (400 MHz, CDCl_3_) δ ppm (*J* Hz) 7.76–7.77 (m, H*^p^*, H*^o^*, 6H), 7.66 (m, H*^m^*, 6H, ^3^*J*_HH_ 7.7, ^4^*J*_PH_ 3.6), 5.84 (ddt, H^2′^, 1H, ^3^*J*_HH_ 17.2, ^3^*J*_HH_ 10.4, ^3^*J*_HH_ 5.6), 5.21 (ddt, H^3′^*_trans_*, 1H, ^3^*J*_HH_ 17.2, ^2^*J*_HH_ 1.2, ^4^*J*_HH_ 1.2), 5.14 (br d, H^3′^*_cis_*, 1H, ^3^*J*_HH_ 10.4, ^2^*J*_HH_ 1.2, ^4^*J*_HH_ 0.9), 4.28 (br m, H^2^, 1H), 3.98–4.0 (m, H^3^, 2H, *AB*-part of *ABX*-spectrum, ^2^*J*_H_^3A^_H_^3B^ 12.8, ^3^*J*_H_^2^_H_^3^ 5.6), 3.74–3.83 (m, H^1B^, H^1′^, 3H, ^2^*J*_PH_^1B^ 10.8, ^2^*J*_H_^1A^_H_^1B^ 15.4, ^3^*J*_H_^2^_H_^1B^ 10.4), 3.67 (ddd, H^1A^, 1H, ^3^*J*_H_^1B^_H_^1A^ 15.4, ^2^*J*_PH_^1A^ 13.2, ^3^*J*_H_^2^_H_^1A^ 2.4); ^13^C NMR spectrum (100.6 MHz, CDCl_3_) δ_C_ ppm (*J* Hz) 134.98 (dtd (d), C*^p^*, ^1^*J*_HC_ 164.1, ^3^*J*_HCCC_ 7.1, ^4^*J*_PCCCC_ 3.1), 134.37 (dtdd (s), C^2′^, ^1^*J*_HC_ 155.3, ^2^*J*_HCC_ 4.5, ^2^*J*_HCC_ 4.5, ^2^*J*_HCC_ 2.8), 134.01 (dddd (d), C*^o^*, ^1^*J*_HC_ 164.6, ^2^*J*_PCC_ 10.3, ^3^*J*_HCCC_ 7.8, ^3^*J*_HCCC_ 6.5), 130.35 (ddd (d), C*^m^*, ^1^*J*_HC_ 165.8, ^3^*J*_PCC_ 12.9, ^3^*J*_HCCC_ 7.3), 118.76 (dt (d), C*^i^*, ^1^*J*_PC_ 87.1, ^3^*J*_HCCC_ 8.5), 117.49 (ddt, C^3′^, ^1^*J*_HC_ 158.8, ^1^*J*_HC_ 155.2, ^3^*J*_HCCC_ 5.5), 73.58 (tdm (d), C^3^, ^1^*J*_HC_ 144.2, ^3^*J*_PCCC_ 14.3), 72.37 ((s), C^1′^, ^1^*J*_HC_ 142.3, ^3^*J*_HCCC_ 11.8, ^3^*J*_HCCC_ 6.9, ^2^*J*_HCC_ 10.3, ^3^*J*_HCOC_ 3.6), 65.24 (dm (d), C^2^, ^1^*J*_HC_ 147.1, ^2^*J*_PCC_ 5.7, ^2^*J*_HCC_ 2.8), 28.54 (tdm (d), C^1^, ^1^*J*_HC_ 131.8, ^1^*J*_PC_ 54.1); ^31^P-{^1^H} NMR spectrum (242.94 MHz, CDCl_3_) δ_P_ 24.2. Found: anal.%: C 57.07; H 5.31; P 6.11, calcd. for C_24_H_26_IO_2_P, %: C 57.16; H 5.20; P 6.14.

**(3-(Decyloxy)-2-hydroxypropyl)triphenylphosphonium iodide** (**4d**). Yield: 81%; mp 75–77 °C; IRS (KBr, cm^−1^): 3297 very br s (OH), 3054, 3023, 2991, 2922, 2853, 1587, 1484, 1459, 1436, 1403, 1369, 1335, 1299, 1234, 1189, 1111, 1026, 996, 943, 877, 834, 793, 746, 716, 689, 533, 508, 494. ^1^H NMR spectrum (400 MHz, CDCl_3_) δ ppm (*J* Hz) 7.39–7.41 (m, H*^p^*, H*^o^*, 9H), 7.28 (m, H*^m^*, 6H, ^3^*J*_HH_ 7.0–7.2, ^4^*J*_PH_ 3.4), 4.28 (very br s OH, 1H), 3.86 (m, H^2^, 1H), 3.41 (ddd, H^1B^, 1H, ^2^*J*_H_^1A^_H_^1B^ 15.3, ^2^*J*_PH_^1B^ 11.1, ^3^*J*_H_^2^_H_^1B^ 10.3), 3.28–3.30 (m, H^1A^, *A*-part of *ABXY*-spectrum; H^3^, *AB*-part of *ABX*-spectrum, 3H), 3.02–3.06 (m, H^1′^, 2H, *AB*-part of *ABX*_2_-spectrum), 1.12 (m, H^3′^, 2H, ^3^*J*_HH_ 6.8–7.0), 0.85–0.88 (m, H^4′^-H^9′^, 14H), 0.46 (t, H^10′^, 3H, ^3^*J*_HH_ 7.1); ^13^C NMR spectrum (100.6 MHz, CDCl_3_) δ_C_ ppm (*J* Hz) 134.30 (dtd (d), C*^p^*, ^1^*J*_HC_ 163.7, ^3^*J*_HCCC_ 7.0, ^4^*J*_PCCCC_ 2.8), 133.37 (dddd (d), C*^o^*, ^1^*J*_HC_ 164.7, ^2^*J*_PCC_ 10.3, ^3^*J*_HCCC_ 7.4, ^3^*J*_HCCC_ 6.7), 129.69 (ddd (d), C*^m^*, ^1^*J*_HC_ 165.8, ^3^*J*_PCCC_ 12.8, ^3^*J*_HCCC_ 7.4), 118.15 (dt (d), C*^i^*, ^1^*J*_PC_ 87.2, ^3^*J*_HCCC_ 8.3), 73.44 (tdm (d), C^3^, ^1^*J*_HC_ 142.8, ^3^*J*_PCCC_ 14.6), 70.99 (br tm (s), C^1′^, ^1^*J*_HC_ 141.0), 64.52 (br dm (d), C^2^, ^1^*J*_HC_ 145.6, ^2^*J*_PCC_ 5.7), 31.26 (tm (s), C^2′^, ^1^*J*_HC_ 125.7), 20.02 (tm (s), C^3′^, C^4′^, ^1^*J*_HC_ 124.4–124.6), 28.95 (tm, (s) C^5′^, ^1^*J*_HC_ 124.0–125.0), 28.87 (tm, (s) C^6′^, ^1^*J*_HC_ 124.0–125.0), 28.70 (tm (s), C^7′^, ^1^*J*_HC_ 124.0–125.0), 27.65 (tdm (d), C^1^, ^1^*J*_HC_ 133.1, ^1^*J*_PC_ 54.2), 25.62 (tm (s), C^8′^, ^1^*J*_HC_ 124.4), 22.03 (tm (s), C^9′^, ^1^*J*_HC_ 125.8), 13.48 (qm (s), C^10′^, ^1^*J*_HC_ 124.2); ^31^P-{^1^H} NMR spectrum (162.0 MHz, CDCl_3_) δ_P_ 25.1 (s);^1^H NMR spectrum (400 MHz, CD_3_CN) δ ppm (*J* Hz) 7.85 (m, H*^p^*, 3H, ^3^*J*_HH_ 7.0–7.2, ^5^*J*_PH_ 2.0, ^4^*J*_HH_ 1.2), 7.82 (m, H*^o^*, 6H, ^3^*J*_PH_ 13.6, ^3^*J*_HH_ 7.0–7.2, ^4^*J*_HH_ 1.3), 7.70 (m, H*^m^*, 6H, ^3^*J*_HH_ 7.0–7.2, ^4^*J*_PH_ 3.5), 4.09 (m, H^2^, 1H), 3.78 (ddd, H^1B^, 1H, ^2^*J*_H_^1A^_H_^1B^ 15.7, ^2^*J*_PH_^1B^ 11.5, ^3^*J*_H_^2^_H_^1B^ 10.2), 3.89 (br s, OH, 1H), 3.56–3.62 (m, H^1A^, H^3^, 3H), 3.42 and 3.45 (two m, H^1′^, 2H, *AB*-part of *ABX*_2_-spectrum, ^2^*J*_H_^1′A^_H_^1′B^ 9.5, ^3^*J*_HH_ 6.6–6.8), 1.53 (m, H^2′^, 2H, ^3^*J*_HH_ 7.0–7.1, ^3^*J*_HH_ 7.0–7.1), 1.27–1.34 (m, H^4′^-H^9′^, 14H), 0.88 (t, H^10′^, 3H, ^3^*J*_HH_ 6.8); ^13^C NMR spectrum (100.6 MHz, CD_3_CN) δ_C_ ppm (*J* Hz) 135.46 (dtm (d), C*^p^*, ^1^*J*_HC_ 164.9, ^3^*J*_HCCC_ 7.3, ^4^*J*_PCCCC_ 3.0, ^2^*J*_HCC_ 1.4, ^2^*J*_HCC_ 0), 134.71 (ddm (d), C*^o^*, ^1^*J*_HC_ 165.2, ^2^*J*_PCC_ 10.3, ^3^*J*_HCCC_ 7.8, ^3^*J*_HCCC_ 7.0), 130.83 (ddd (d), C*^m^*, ^1^*J*_HC_ 166.2, ^3^*J*_PCCC_ 12.8, ^3^*J*_HCCC_7.3), 119.97 (dt (d), C*^i^*, ^1^*J*_PC_ 87.4, ^3^*J*_HCCC_ 7.4), 74.87 (tdm (d), C^3^, ^1^*J*_HC_ 142.0, ^3^*J*_PCCC_ 14.8), 71.96 (tm (s), C^1′^, ^1^*J*_HC_ 140.1), 65.64 (dm (d), C^2^, ^1^*J*_HC_ 145.6, ^2^*J*_PCC_ 5.8, ^2^*J*_HCC_ 2.8, ^2^*J*_HCC_ 2.8), 32.34 (tm (s), C^2′^, ^1^*J*_HC_ 125.3), 30.13 (tm (s), C^3′^, ^1^*J*_HC_ 124.0–125.0), 30.10 (tm (s), C^4′^, ^1^*J*_HC_ 124.0–125.0), 30.03 (tm (s), C^5′^, ^1^*J*_HC_ 124.0–125.0), 29.93 (tm (s), C^6′^, ^1^*J*_HC_ 124.0–125.0), 29.77 (tm (s), C^7′^, ^1^*J*_HC_ 124.0–125.0), 28.09 (tdm (d), C^1^, ^1^*J*_HC_ 134.0, ^1^*J*_PC_ 54.6), 26.61 (tm (s), C^8′^, ^1^*J*_HC_ 124.4), 23.11 (tm (s), C^9′^, ^1^*J*_HC_ 124.5), 14.22 (qm (s), C^10′^, ^1^*J*_HC_ 124.4, ^3^*J*_HCCC_ 3.5–4.0, ^2^*J*_HCC_ 3.3–3.4); ^31^P-{^1^H} NMR spectrum (162.0 MHz, CD_3_CN) δ_P_ 24.6 (s); MALDI-MS, *m*/*z* 477.6 [M − I]^+^. Calcd. for C_31_H_42_O_2_P^+^: 477.7. Found: anal. C 62.03; H 6.87; I 20.94%, calcd. for C_31_H_42_IO_2_P, C 61.59; H 7.00; I 20.99%.

#### 3.2.6. Synthesis of **3d** and **7**

To a solution of the corresponding previously obtained compound **3a** (1.5 g, 3 mmol) in CH_3_CN (5 mL) was added NaI (1.8 g, 12 mmol) and 10% 18-crown-6 (0.08 g). The reaction mixture was stirredunder reflux for 36 (h). After cooling the mixture to room temperature, the precipitate was remover and the solvent was evaporatedin *vacuo*. The residue was purified by chromatography on silica gel using a EtOAc as eluent to afford the desired product **3d** and **7**.

**(2-Hydroxy-3-iodopropyl)triphenylphosphonium trifluoromethanesulfonate** (**3d**). Yield: 31%; *R_f_* = 0.88 (silica gel, CH_3_CN); mp 178–180 °C; IRS (KBr, cm^−1^): 3219 (OH), 2869, 1438, 1407, 1251, 1180, 1109, 1040, 769, 752, 721, 692, 653, 580, 539, 521, 499, 475; ^1^H NMR spectrum (400 MHz, CD_3_CN) δ ppm (*J* Hz) 7.88 (m, H*^p^*, 3H, ^3^*J*_HH_ 7.2, ^5^*J*_PH_ 1.8, ^4^*J*_HH_ 1.7), 7.78 (m, H*^o^*, 6H, ^3^*J*_PH_ 12.8, ^3^*J*_HH_ 7.2, ^4^*J*_HH_ 1.7), 7.73 (m, H*^m^*, 6H, ^3^*J*_HH_ 7.2, ^3^*J*_HH_ 7.2, ^4^*J*_PH_ 3.7), 4.24 (br d, OH, 1H, ^3^*J*_HH_ 4.0), 3.84 (m, H^2^, 1H), 3.70 (ddd, H^1B^, 1H, *B*-part of *ABX*-spectrum, ^2^*J*_H_^1A^_H_^1B^ 15.2, ^2^*J*_PH_^1B^ 11.5, ^3^*J*_H2H1B_ 10.5), 3.50–3.56 (m, H^1A^, H^3^, 3H); ^13^C NMR spectrum (150.9 MHz, CD_3_CN, δ_C_ ppm, *J* Hz): 135.92 (dtm (d), C*^p^*, ^1^*J*_HC_ 161.7, ^3^*J*_HCCC_ 7.3, ^4^*J*_PCCCC_ 3.0, ^2^*J*_HCC_ 1.8), 135.02 (dddd (d), C*^o^*, ^1^*J*_HC_ 165.1, ^2^*J*_PCC_ 10.4, ^3^*J*_HCCC_ 7.6, ^3^*J*_HCCC_ 6.5), 131.08 (dddd (d), C*^m^*, ^1^*J*_HC_ 164.7–165.5, ^3^*J*_PCCC_ 12.8, ^3^*J*_HCCC_ 7.3–7.5, ^2^*J*_HCC_ 1.8–2.4), 121.95 (q (q), CF_3_, ^1^*J*_FC_ 320.2), 120.10 (ddd (d), C*^i^*, ^1^*J*_PC_ 87.6, ^3^*J*_HCCC_ 8.0, ^3^*J*_HCCC_ 7.1), 66.41 (dm (d), C^2^, ^1^*J*_HC_ 142.2, ^3^*J*_PCCC_ 3.4), 31.35 (tdm (d), C^1^, ^1^*J*_HC_ 132.1, ^1^*J*_PC_ 54.5, ^2^*J*_HCC_ 4.2, ^3^*J*_HCCC_ 2.0–2.2), 15.06 (tdm (d), C^3^, ^1^*J*_HC_ 152.8, ^3^*J*_PCCC_ 17.4, ^2^*J*_HCC_ 3.6, ^3^*J*_HCCC_ 2.3); ^19^F NMR spectrum (376.5 MHz, CDCl_3_) δ_F_ −78.27 (s); ^31^P-{^1^H} NMR spectrum (242.94 MHz, CD_3_CN) δ_P_ 23.3 (s); MALDI-MS, *m*/*z* 447.1 [M − CF_3_SO_3_]^+^. Calcd. for C_21_H_21_IOP^+^: 447.3. Found: anal. C 44.26; H 3.33; I 21.25%, calcd. for C_22_H_21_F_3_IO_4_PS, C 44.31; H 3.55, I 21.28%.

**(2-Hydroxy-3-iodopropyl)triphenylphosphonium iodide** (**7**). Yield: 56%; *R_f_* = 0.69 (silica gel, CH_3_CN); mp 197–198 °C; IRS (KBr, cm^−1^): 3399 br s, 3064, 2996, 2960, 2923, 2910, 1588, 1486, 1459, 1439, 1416, 1382, 1337, 1282, 1249, 1225, 1196, 1161, 1110, 1046, 1026, 997, 969, 949, 869, 849, 843, 792, 750, 725, 715, 691, 638, 573, 540, 510, 497, 474. ^1^H NMR spectrum (400 MHz, CD_3_CN) δ ppm (*J* Hz) 7.76–7.79 (br m, H*^p^*, H*^o^*, 9H, ^3^*J*_HH_ 7.1–7.2, ^3^*J*_PH_ 12.5), 7.68 (m, H*^m^*, 6H, ^3^*J*_HH_ 7.1–7.2, ^3^*J*_HH_ 7.1–7.2, ^4^*J*_PH_ 3.3), 4.62 (br s, OH, 1H), 4.00 (m, H^2^, 1H), 3.79–4.85 (m, H^1^, 2H), 3.61 (m, H^3^, 2H); ^13^C NMR spectrum (100.6 MHz, CDCl_3_ + 20% DMCO-*d*_6_) δ ppm (*J* Hz) 133.55 (dtd (d), C*^p^*, ^1^*J*_HC_ 164.2, ^3^*J*_HCCC_ 7.2, ^4^*J*_PCCCC_ 2.6), 132.66 (dddd (d), C*^o^*, ^1^*J*_HC_ 165.1, ^2^*J*_PCC_ 10.4, ^3^*J*_HCCC_ 7.2, ^3^*J*_HCCC_ 6.7), 128.78 (ddd (d), C*^m^*, ^1^*J*_HC_ 165.9, ^3^*J*_PCCC_ 12.8, ^3^*J*_HCCC_ 7.4), 117.72 (ddd (d), C*^i^*, ^1^*J*_PC_ 87.5, ^3^*J*_HCCC_ 9.0), 63.76 (dm (d), C^2^, ^1^*J*_HC_ 149.5, ^2^*J*_PCC_ 5.0), 29.58 (tdm (d), C^1^, ^1^*J*_HC_ 131.0, ^1^*J*_PC_ 53.7), 14.56 (tdm (d), C^3^, ^1^*J*_HC_ 151.5, ^3^*J*_PCCC_ 17.8, ^2^*J*_HCC_ 4.3).^13^C NMR spectrum (150.9 MHz, CDCl_3_ + 10% MeCN) δ_C_ ppm (*J* Hz) 135.13 (d. t. d (d), C*^p^*, ^1^*J*_HC_ 165.1, ^3^*J*_HCCC_ 7.2, ^4^*J*_PCCCC_ 2.9), 134.16 (dddd (d), C*^o^*, ^1^*J*_HC_ 165.0, ^2^*J*_PCC_ 10.4, ^3^*J*_HCCC_ 7.6, ^3^*J*_HCCC_ 6.7), 130.46 (ddd (d), C*^m^*, ^1^*J*_HC_ 166.9, ^3^*J*_PCCC_ 12.7, ^3^*J*_HCCC_ 7.2), 118.59 (ddd (d), C*^i^*, ^1^*J*_PC_ 87.2, ^3^*J*_HCCC_ 8.4), 65.72 (dm (d), C^2^, ^1^*J*_HC_ 148.5, ^2^*J*_PCC_ 4.4), 31.10 (tdm (d), C^1^, ^1^*J*_HC_ 130.1, ^1^*J*_PC_ 54.6), 14.56 (tdm (d), C^3^, ^1^*J*_HC_ 152.1, ^3^*J*_PCCC_ 16.9, ^2^*J*_HCC_ 3.4).^31^P-{^1^H} NMR spectrum (242.94 MHz, CD_3_CN) δ_P_ 23.9 (s); MALDI-MS, *m*/*z* 447.4[M − I]^+^. Calcd. for C_21_H_21_IOP^+^: 447.3. Found: anal. C 43.72; H 3.62; I 44.10%, calcd. for C_21_H_21_I_2_OP, C 43.93; H 3.69; I 44.20%.

#### 3.2.7. Preparation and Characterization Phosphorus-Containing Amphiphile Liposomes

##### Chemicals

L-α-phosphatidylcholine (PC) (Soy, 95%, Avanti polar lipids), rhodamine B (99%, ACROS Organics, NJ, USA), Ultra-purified water (18.2 MΩcm resistivity at 25 °C) was produced from Direct-Q 5 UV equipment (Millipore S.A.S. 67120 Molsheim, France). All reagents were used without further treatment.

##### Preparation and Characterization

L-α-phosphatidylcholine (PC) and **3k**,**l** (5% *w*/*w*) were dissolved in 1 mL of ethanol. The homogeneous solution was kept in a water bath at 60 °C until alcohol evaporation to obtain a thin lipid film. Ultra-purified waterwas pre-heated to 60 °C and added to rehydrate the lipids at 60 °C in the absence or presence of Rhodamine B (0.1% *w*/*w*). The solution was stirred under magnetic stirring (750 rpm) (Heidolph, Germany) for 30 min at the same temperature. Then the solution was kept for 1.5 h in a water bath at 37 °C. The multilamellar liposomes were extruded 15 times by passage through a polycarbonate membrane of 100 nm pore size (Mini-Extruder Extrusion Technique, Avanti Polar Lipids, Inc., Birmingham, AL, USA).

The mean particle size, zeta potential, and polydispersity index were determined by dynamic light scattering (DLS), using a Malvern Instrument Zetasizer Nano (Malvern, Worcestershire, UK) and Litesizer 500 Anton Paar (Anto Paar Graz, Graz, Austria). The size (hydrodynamic diameter, nm) was calculated according to the Einstein–Stokes relationship *D = kBT*/*3πηx*, in which *D* is the diffusion coefficient, *k_B_* is the Boltzmann’s constant, *T* is the absolute temperature, *η* is the viscosity, and *x* is the hydrodynamic diameter of the nanoparticles. The diffusion coefficient was determined at least in triplicate for each sample. The average error of the measurements was approximately 10%. All samples were diluted (20 times)with ultra-purified water to a suitable concentration (2.5 mg/mL) and analyzed in triplicate.

##### In Vitro Rhodamine B Release Profile

The monitoring of rhodamine B release from liposomes was performed using the dialysis bag diffusion method. Dialysis bags retain liposomes and allow the released rhodamine B to diffuse into the medium. The bags were soaked in Milli-Q water for 12 h before use. Then, 0.4 mL of liposomes were poured into the dialysis bag. The two bag ends were sealed with clamps. The bags were then placed in a vessel containing 100 mL of 0.025 M sodium phosphate buffer pH 7.4, the receiving phase. The vessel was placed in a thermostatic shaker (New Brunswick, NJ, USA) at 37 °C, under a stirring rate of 150 rpm. At predetermined time intervals, 0.5 mL of samples were withdrawn, and their absorbance at 554 nm was measured using Perkin Elmer λ35 (Perkin Elmer Instruments, Norwalk, CT, USA). All samples were analyzed in triplicate. The extinction coefficient of rhodamine B is 106,089 M^−1^ cm^−1^ at pH = 7.4.

##### Encapsulation Efficiency and Loading Capacity

Encapsulation efficiency (EE, %) and loading capacity (LC, %) were assessed for samples containing rhodamine B. These parameters were determined indirectly by filtration/centrifugation technique, measuring free rhodamine B (non-encapsulated) by spectrophotometry. A volume of 50 µL of each rhodamine B-loaded liposomes was placed in centrifugal filter devices Nanosep centrifugal device 3K Omega (Pall Corporation) to separate lipid and aqueous phases and centrifuged at 10,000 rpm for 30 min using centrifuge MiniSpin plus (Eppendorf AG, Hamburg, Germany). Free rhodamine B was quantified by UV absorbance using PerkinElmer λ35 (Perkin Elmer Instruments, Waltham, MA, USA) at 554 nm (ε = 106,089 M^−1^ cm^−1^ in 0.0025 M phosphate buffer at pH = 7.4). The encapsulation parameters were calculated against appropriate calibration curve, using the following equation:(1)$$\mathrm{EE}\left(\%\right)=\frac{\mathrm{Total}\;\mathrm{amount}\;\mathrm{of}\;\mathrm{RhodB}-\mathrm{Free}\;\mathrm{RhodB}}{\mathrm{Total}\;\mathrm{amount}\;\mathrm{of}\;\mathrm{RhodB}}\times 100\%$$
(2)$$\mathrm{LC}\left(\%\right)=\frac{\mathrm{Total}\;\mathrm{amount}\;\mathrm{of}\;\mathrm{RhodB}-\mathrm{Free}\;\mathrm{RhodB}}{\mathrm{Total}\;\mathrm{amount}\;\mathrm{of}\;\mathrm{phospholipid}}\times 100\%$$

### 3.3. Biology

#### 3.3.1. Cell Toxicity Assay (MTT-Test)

The toxic effect on cells was determined using the colorimetric method of cell proliferation MTT (Thiazolyl Blue Tetrazolium Bromide, Sigma). For this, 10 μL of MTT reagent in Hank’s balanced salt solution (HBSS) (final concentration 0.5 mg/mL) was added to each well. The plates were incubated at 37 °C for 2–3 h in an atmosphere humidified with 5% CO_2_. Absorbance was recorded at 540 nm using an Invitrologic microplate reader (Russia). Experiments for all compounds were repeated three times. The M-HeLa clone 11 human, epithelioid cervical carcinoma, strain of HeLa, clone of M– HeLa; human alveolar adenocarcinoma cells (A549); human duodenal cancer cell line (HuTu 80); human breast adenocarcinoma cells (MCF-7); glioblastoma cell line (T98G); Wi-38 VA-13 cell culture, subline 2RA (human embryonic lung) from the Type Culture Collection of the Institute of Cytology (Russian Academy of Sciences) and PC-3 human Caucasian prostate adenocarcinoma from Type Culture Collection (ATCC, Manassas, VA, USA) were used in the experiments. The cells were cultured on a standard nutrient medium “Igla” produced by the Moscow Institute of Poliomyelitis and Viral Encephalitis. M.P.Chumakov by PanEco with the addition of 10% fetal calf serum and 1% nonessential amino acids (NEAA).

The cells were sown on a 96-well panel from Eppendorf at a concentration of 5 × 103 cells per well in a volume of 100 μL of medium and cultured in a CO_2_ incubator at 37 °C. In 48 h after planting the cells, the culture medium was taken into the wells, and 100 μL of solutions of the studied drug in the specified dilutions were added to the wells. Dilutions of the compounds were prepared directly in growth medium supplemented with 5% DMSO to improve solubility. The cytotoxic effect of the test compounds was determined at concentrations of 0.1–100 μM. The calculation of the IC_50_, the concentration of the drug causing inhibition of cell growth by 50%, was performed using the program: MLA−“Quest Graph ™ IC_50_ Calculator.” AAT Bioquest, Inc., https://www.aatbio.com/tools/ic50-calculator, accessed on 25 June 2021.

#### 3.3.2. Induction of Apoptotic Effects by Test Compounds (Flow Cytometry Assay)

##### Cell Culture

HuTu 80 cells at 1 × 10^6^ cells/well in a final volume of 2 mL were seeded into six-well plates. After 48 h of incubation, various concentrations of compounds **3a**,**b** and **3k** were added to wells.

##### Cell Apoptosis Analysis

The cells were harvested at 2000 rpm for 5 min and, then, washed twice with ice-cold PBS, followed by resuspension in binding buffer. Next, the samples were incubated with 5 μL of annexin V-Alexa Fluor 647 (Sigma–Aldrich, St. Louis, MO, USA) and 5 μL of propidium iodide for 15 min at room temperature in the dark. Finally, the cells were analyzed by flow cytometry (Guava easy Cyte, MERCK, Rahway, NJ, USA) within 1 h. The experiments were repeated three times.

##### Mitochondrial Membrane Potential

Cells were harvested at 2000 rpm for 5 min and then washed twice with ice-cold PBS, followed by resuspension in JC-10 (10 µg/mL) and incubation at 37 °C for 10 min. After the cells were rinsed three times and suspended in PBS, the JC-10 fluorescence was observed by flow cytometry (Guava easy Cyte, MERCK, NJ, USA).

##### Detection of Intracellular ROS

HuTu 80 cells were incubated with compounds **3a**,**b** and **3k** at concentrations of IC_50_/2 and IC_50_ for 48 h. ROS generation was investigated using flow cytometry assay and CellROX^®^ Deep Red flow cytometry kit. For this HuTu 80 cells were harvested at 2000 rpm for 5 min and then washed twice with ice-cold PBS, followed by resuspension in 0.1 mL of medium without FBS, to which was added 0.2 μL of Cell ROX^®^ Deep Red and incubated at 37 °C for 30 min After three times washing the cells and suspending them in PBS, the production of ROS in the cells was immediately monitored using flow cytometer Guava easy Cyte, MERCK, NJ, USA).

##### Statistical Analysis

The IC_50_ values were calculated using the online calculator MLA−Quest Graph™ IC_50_ Calculator AAT Bioquest, Inc., 25 June 2021. Statistical analysis was performed using the Mann–Whitney test (*p* < 0.05). Tabular and graphical data contains averages and standard error.

## 4. Conclusions

A mild and effective approach to the synthesis of the functionally substituted (2-hydroxypropyl) triphenylphosphonium triflates is chemoselective reaction of triphenylphosphonium triflate with halomethyloxiranes and glycidyl ethers, proceeding by the S_N_2 mechanism. Another convenient method for the 3-alkoxy-(2-hydroxypropyl) triphenylphosphonium iodides synthesis is based on the reaction of 3-alkoxy-2-hydroxyiodopropane with triphenylphosphine. All molecular and crystal structures were corroborated by NMR and XRD.

All obtained phosphonium salts exhibit from moderate to high cytotoxicity against human cancer cell lines M-HeLa, MCF-7, A549, HuTu-80, PC3, T98G. The cytotoxic activity of the most active compounds **3a**, **b**, **k**, **j** and **4c** is caused by the induction of apoptosis via the mitochondrial ROS pathway. Improving the bioavailability, cytotoxicity, stability, and reducing the toxicity of phosphonium salts was achieved by using a nanotechnological approach based on liposomal systems.

The development of functionalized 2-hydroxypropylphosphonium salts is promising for creation of the new effective mitochondria-targeted anticancer agents. The presence a hydroxyl group will allow further introduction of the additional pharmacophore fragments.

## Data Availability

The data presented in this study are available on request from the corresponding author.

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
