# Peer review of "Rational Design 2-Hydroxypropylphosphonium Salts as Cancer Cell Mitochondria-Targeted Vectors: Synthesis, Structure, and Biological Properties"

_molecules, 2021, doi:10.3390/molecules26216350_

Round 1
Reviewer 1 Report
The manuscript presents some interesting research on (2-hydroxypropyl)triphenylphosphonium triflates and iodides substituted in the 3-position with an alkoxy, alkylcarboxyl group, or halogen, which display antitumor activity. The authors obtained these compounds by presenting an efficient synthesis method and characterized by defining the exact structure using research techniques such as NMR, MALDI-MS, XRD – all confirmed by attached copies of NMR spectra and x-ray diffraction data for the crystals in supporting information.
Cytotoxicity of these salts against six cancer and one normal cell lines revealed two most active compounds with the high selectivity of the action.
Additionally, for three among syntesized salts the apoptosis induction was studied and two other phosphonium salts were tested as modifiers of phospholipidic liposomes. I think that the authors should justify the selection of individual specific phosphonium salts for these studies, because there is no presented reason for such a selection of salts.
Other comments to consideration by the authors:
- Insert into the text on the page 3 a reference to the Scheme 4
- Correct the notation of the oxirane bridge and the lactone in the reaction equations in Schemes 3 and 4 - the bracket with the subscript „n” should be outside the ring; the oxiran bridge should connect two adjacent carbon atoms
- On page 4, line 129, replace „diphenylphosphine” with „triphenylphosphine”
- Z (mV) data in Table 2 should be reported with the same accuracy, one digit after the decimal point; discussion of the results for the data contained in Table 2 should be more precise and accurately include the data from the table
Author Response
Dear reviewer, please find the attached file to see the point-by-point response .

Reviewer 2 Report
The manuscript entitled “Rational Design 2-Hydroxypropylphosphonium Salts as Cancer 2 Cell Mitochondria-targeted Vectors. Synthesis, Structure and Biological Properties” was submitted by Vladimir F. Mironov et al presents a mild and effective approach for the synthesis of functionally substituted (2-hydroxypropyl) triphenylphosphonium triflates. The structures of products were confirmed by 1H 13C and 31P NMR, IR spectra, element analysis and XRD data. Obtained products were screened as anticancer agents on 6 cell lines. Anticancer activity was compared with Doxorubicin. It was demonstrated that functionalized 2-hydroxypropylphosphonium salts is promising for the creation of new effective mitochondria-targeted anticancer agents. The presence of a hydroxyl group will allow the further introduction of additional pharmacophore fragments.
Thus, I would like to recommend the manuscript for publication in Molecules. However, there are some minor issues needed to be addressed before publication.
Minor issues:
2.1.4. Liposomal Systems Based on Amphiphilic Triflates of Acyloxypropylphosphonium and L-α-Phosphatidylcholine. What is the rationale for Rhodamine B encapsulation experiments?
In Table 1. Compounds 3k and 3l are cationic amphiphiles they could form also form nanoparticles. These data could be included in this article.
Table 2 and Table 3 are duplicated results. These tables should emerge.
Paragraph 2.2.2. there were selected 3a,3b and 3k as leading compounds but not selected 3j, 4b and 4c. What is the justification for such a choice?
Materials and Methods. TLC was used for process monitoring. I could not find any information in the article regarding retention factors and eluent systems.
Spectra should be presented in a uniform style. If coupling constants are given then m – multiples should be changed to multiplicity of signal (s, d, t or others)
1-Chloro-3-methoxypropan-2-ol (5a) 1H-NMR. Missing protons.
1-Chloro-3-(decyloxy)propan-2-ol (5d). Missing protons.
3-(Methoxymethyl)oxirane (2f). Missing protons.
3-((Allyloxy)methyl)oxirane (2i). Number of protons? 3.12 (m, H2, 3JH3AH2 5.8, 3JH1MH2 4.1, 3JH3MH2 3.0, 3JH1AH2 2.7).
1-Ethoxy-3-iodopropan-2-ol (6b). Missing protons.
1-(Decyloxy)-3-iodopropan-2-ol (6d). B.p. are in Torr but in all other cases it was in mmHg.
(3-Bromo-2-hydroxypropyl)triphenylphosphonium trifluoromethane sulfonate. Missing protons.
(2-Hydroxy-3-methoxypropyl)triphenylphosphonium trifluoromethanesul fonate (3f). Coupling constants should be given as numbers. Not interval. 7.67 (m, Ho, 6H, 3JPH 12.5, 3JHH 7.1-7.2, 4JHH 1.5),
(3-(Allyloxy)-2-hydroxypropyl)triphenylphosphonium trifluoromethanesul fonate (3i). 4.10 (br m, H2, 712 OH, 2H). the compound has only one OH group.
(3-(Decyloxy)-2-hydroxypropyl)triphenylphosphonium trifluoromethanesul fonate (3j). The compound has only one OH group.
(2-Hydroxy-3-palmitoyloxy-propyl)triphenylphosphonium trifluoro methanesulfonate (3k). Missing protons.
(2-Hydroxy-3-stearoyloxy-propyl)triphenylphosphonium trifluoromethanesul fonate (3l). Missing protons.
(3-Ethoxy-2-hydroxypropyl)triphenylphosphonium iodide (4b). Missing protons.
(3-(Allyloxy)-2-hydroxypropyl)triphenylphosphonium iodide (4c). Missing protons.
3.2.7.1. Preparation and characterization. What was the final concentration of analyzed samples?
Author Response

(The authors gave the same response as above.)

Reviewer 3 Report
The reviewed paper concerns the synthesis and cytotoxic activity of 2-hydroxypropylphosphonium salts. The strongest side of the work is the experimental (synthetic) part, especially the high purity of compounds obtained (the NMR spectra show practically no impurities, which is impressive for the synthesis of phosphonium salts). However, the presentation of the results require some corrections:
- Scheme 4: I suggest you number the reaction equations and then give specific references in the text (e.g. scheme 4/a, scheme 4/b, or something like that).
- Scheme 5: I do not really understand the signatures 2e (R, S) and 3e (R, S); in compounds 2 and 3 I do not see two stereogenic centers.
- Page 4, lines 139-140: The sentence: “The salt signal 1…” sounds awkward
- Line 323: should not it be: Cytotoxicity against cancer and normal cell lines?
- Table 2; lines 328 and 331: The experiments… - you only need to provide this information once.
- Line 340: Please check “the range 1.1-3.7”
- why for the tests described in paragraph 2.2.2 compounds 3a, b, k were used, since you wrote that the compounds 3j and 4c (see paragraph 2.2.1) showed the highest activity?
- Consider change “at boiling” to “under reflux” in the experimental part
- Conclusions: see the note regarding compounds 3a, b, k vs 3j, 4c; besides, there are a lot of typos and minor linguistic errors – please check it.
- The descriptions of the spectroscopic data are very detailed; in my opinion, a bit too much (the coupling constants for multiplets; descriptions of 13C NMR without decoupling, NMR data in two different solvents...). I propose to simplify them.
- For multiplets please give the range (e.g. 3.98-4.02 instead of 4.00 ppm)
- Compound 3a, line 587: it should be 3.70-3.71 (m, H3, 2H)
- For the IR spectra, please select only the most important bands.
- For compounds: 5a, 5b – lack of IR
- For compounds: 2f-I, 6a-d – lack of 13C NMR
- For compound 6c – lack of b.p.
- I noticed an inconsistent description of the MS data – see lines 781-782 and 825
- Supporting information: please check the descriptions of the spectra, especially the type of solvent used - in many places it does not coincide with the signals on the spectra (Figure 30 (CH2CL2?), Figure 35-44, 51-59), or they are different depending on the experiment (different for 1 HNMR and different for 31P NMR; see figures 129, 137, 145, 153) - in general, please check the solvents.
- Supporting information: please arrange the spectra of compounds 3 in the correct order (see pages 35-39)
- Supporting information: structural formulas will facilitate the analysis of the spectra - consider adding them.
Questions:
- What about the stability of triphenylphosphonium triflate (HPPh3+ OTf-) - have you tried to isolate it, what about other commercially available triphenylphosphonium salts (bromide or tetrafluoroborate; HPPh3+ Br-. HPPh3+ BF4-) - did you try the reaction in their presence?
- Do you have any knowledge about the influence of a type of anion on the biological activity of phosphonium compounds? Have anions like triflate, bromide, or iodide the prospect of being used in therapeutic agents (as far as I know, it is assumed that chlorides are the safest solution here)?
All in all, I think the article is suitable for publication in Molecules after minor corrections.
Author Response

(The authors gave the same response as above.)
